# Ventrolateral periaqueductal gray neurons prioritize threat probability over fear output

Kristina M Wright*, Michael A McDannald*

Psychology Department, Boston College, Chestnut Hill, United States

**Abstract** Faced with potential harm, individuals must estimate the probability of threat and initiate an appropriate fear response. In the prevailing view, threat probability estimates are relayed to the ventrolateral periaqueductal gray (vlPAG) to organize fear output. A straightforward prediction is that vlPAG single-unit activity reflects fear output, invariant of threat probability. We recorded vlPAG single-unit activity in male, Long Evans rats undergoing fear discrimination. Three 10 s auditory cues predicted unique foot shock probabilities: danger (p=1.00), uncertainty (p=0.375) and safety (p=0.00). Fear output was measured by suppression of reward seeking over the entire cue and in one-second cue intervals. Cued fear non-linearly scaled to threat probability and cue-responsive vlPAG single-units scaled their firing on one of two timescales: at onset or ramping toward shock delivery. VlPAG onset activity reflected threat probability, invariant of fear output, while ramping activity reflected both signals with threat probability prioritized.
DOI: https://doi.org/10.7554/eLife.45013.001

*For correspondence:
wrightko@bc.edu (KMW);
michael.mcdannald@bc.edu
(MAM)

**Competing interests:** The authors declare that no competing interests exist.

## Introduction

When confronted with potential harm, an estimate of threat probability must be made and followed by an appropriate fear response. The ventrolateral periaqueductal gray (vlPAG) has long been implicated in this fear response (*Fanselow, 1993*; *Kim et al., 1993*; *Liebman et al., 1970*; *Carrive et al., 1997*; *Bandler et al., 1985*; *Vianna et al., 2001*; *Arico et al., 2017*; *Assareh et al., 2017*). In the canonical fear circuit, threat probability estimates (the stored associative strength of cue and foot shock, for example a danger cue in Pavlovian fear conditioning) originate in amygdalar nuclei (*Duvarci and Pare, 2014*; *Fanselow and LeDoux, 1999*; *Davis, 2006*; *Maren et al., 2013*). Amygdalar threat estimates are sent to the vlPAG, which in turn organizes behavioral components of fear output, most notably freezing (*Perusini and Fanselow, 2015*; *Tovote et al., 2015*; *Dejean et al., 2015*; *Koutsikou et al., 2014*; *Walker et al., 1997*). While the vlPAG contributes to diverse fear and pain-related processes (*Assareh et al., 2017*; *Bellgowan and Helmstetter, 1998*; *Parsons et al., 2010*; *McNally et al., 2011*; *Johansen et al., 2010*), the view of the vlPAG as a node for fear output is so prevalent, it is commonly presented in introductory neuroscience textbooks (*Bear et al., 2016*; *Carlson and Birkett, 2017*).

A series of studies have uncovered a vlPAG population showing short-latency increases in firing to a danger cue. This characteristic would be expected of neurons organizing fear output. Yet in these studies, robust relationships between vlPAG single-unit activity and freezing were observed in a minority of neurons (*Tovote et al., 2016*), weakly observed at danger cue onset (*Watson et al., 2016*), or mixed with activity that purely reflected the danger cue (*Ozawa et al., 2017*). At best, freezing only partially accounted for vlPAG activity. If not freezing, then what aspect of fear do vlPAG neurons signal? Here we test the hypothesis that vlPAG neurons signal threat probability.

We drew from learning theory (*Rescorla, 1968*) to devise a fear discrimination procedure in which three cues predict unique foot shock probabilities: danger (p=1.00), uncertainty (p=0.375) and safety

**eLife digest** The brain is hard-wired to detect and respond to threats. Catching sight of a spider or a snake, or hearing an unfamiliar sound in the house at night, can make you freeze momentarily. Multiple regions of the brain contribute to this process. According to the textbook view of threat detection, the amygdala and prefrontal cortex estimate the size of the threat. They then send this information to an area called the ventrolateral periaqueductal gray (vlPAG). The vlPAG responds by triggering a fear response such as freezing.

But some studies suggest that the vlPAG may do more than just trigger a fear response. To test this idea, Wright and McDannald trained rats to associate three different 10-second tones with different probabilities of receiving a mild electric shock to the foot. One of the tones was followed by a foot shock 100% of the time. Another tone was followed by a foot shock 37.5% of the time, while the third was never followed by a foot shock. The first tone thus signaled danger, the second uncertainty, and the third safety.

Wright and McDannald recorded vlPAG activity as the rats heard each of the tones. The recordings revealed two distinct groups of vlPAG neurons. Both groups responded most to the tone that signaled danger, less to the tone that signaled uncertainty, and least to the tone that signaled safety. However, they responded at different times. One group of neurons was most active at the start of the tone. The activity of this group depended upon the degree of threat, and not upon whether the rat showed a fear response. The second group of neurons increased its activity over the course of each tone. The activity of this group mainly reflected the degree of threat, but also represented the rat's fear response to a lesser extent.

The vlPAG thus helps to signal the size of a threat, rather than simply generating a fear response. This distinction is important because people with anxiety disorders tend to overestimate threats, and many treatments for anxiety target the brain regions involved in threat estimation. Future studies should examine how the vlPAG works together with other areas, including the amygdala and the prefrontal cortex, to evaluate threats. Understanding this circuit in full could ultimately lead to better treatments for phobias and anxiety.
DOI: https://doi.org/10.7554/eLife.45013.002

(p=0.00). This behavioral approach was essential as previous studies utilized procedures in which a single cue predicted foot shock with certainty (*Tovote et al., 2016*; *Watson et al., 2016*; *Ozawa et al., 2017*), precluding the ability to observe neural activity reflecting threat probability. We measured fear output using conditioned suppression of reward seeking (*Rescorla, 1968*; *Bouton and Bolles, 1980*) over the entire cue and in one-second cue intervals. Using this procedure, we have found that suppression non-linearly scales to shock probability. So while suppression is strong to danger, intermediate to uncertainty and weak to safety; uncertainty produces more suppression than would be expected given its shock probability (*Walker et al., 2018*; *Ray et al., 2018*; *DiLeo et al., 2016*; *Wright et al., 2015*; *Berg et al., 2014*). Concurrent with fear discrimination, we recorded vlPAG single-unit activity. Indeed, the vlPAG could potentially signal fear output via conditioned suppression of reward seeking (*Arico et al., 2017*), a long-established measure of fear (*Estes and Skinner, 1941*) that correlates with freezing (*Bouton and Bolles, 1980*). The non-linear relationship between behavior and shock probability permitted us to determine whether vlPAG single-unit activity was better captured by fear output or threat probability.

## Results

Six male, Long Evans rats were trained to nose poke in a central port in order to receive a food pellet from a cup below. During fear discrimination (*Figure 1A*), three distinct auditory cues predicted unique foot shock probabilities: danger (p=1.00), uncertainty (p=0.375) and safety (p=0.00). Trial order was randomized for each rat, each session. Fear was measured with suppression ratio and was calculated by comparing nose poke rates during baseline and cue periods (see Materials and methods). After eight discrimination sessions, rats were implanted with 16-wire, drivable microelectrode bundles dorsal to the vlPAG (*Figure 1B*). Following recovery, rats were returned to fear discrimination. Single-units were isolated and held for the duration of each recording session.

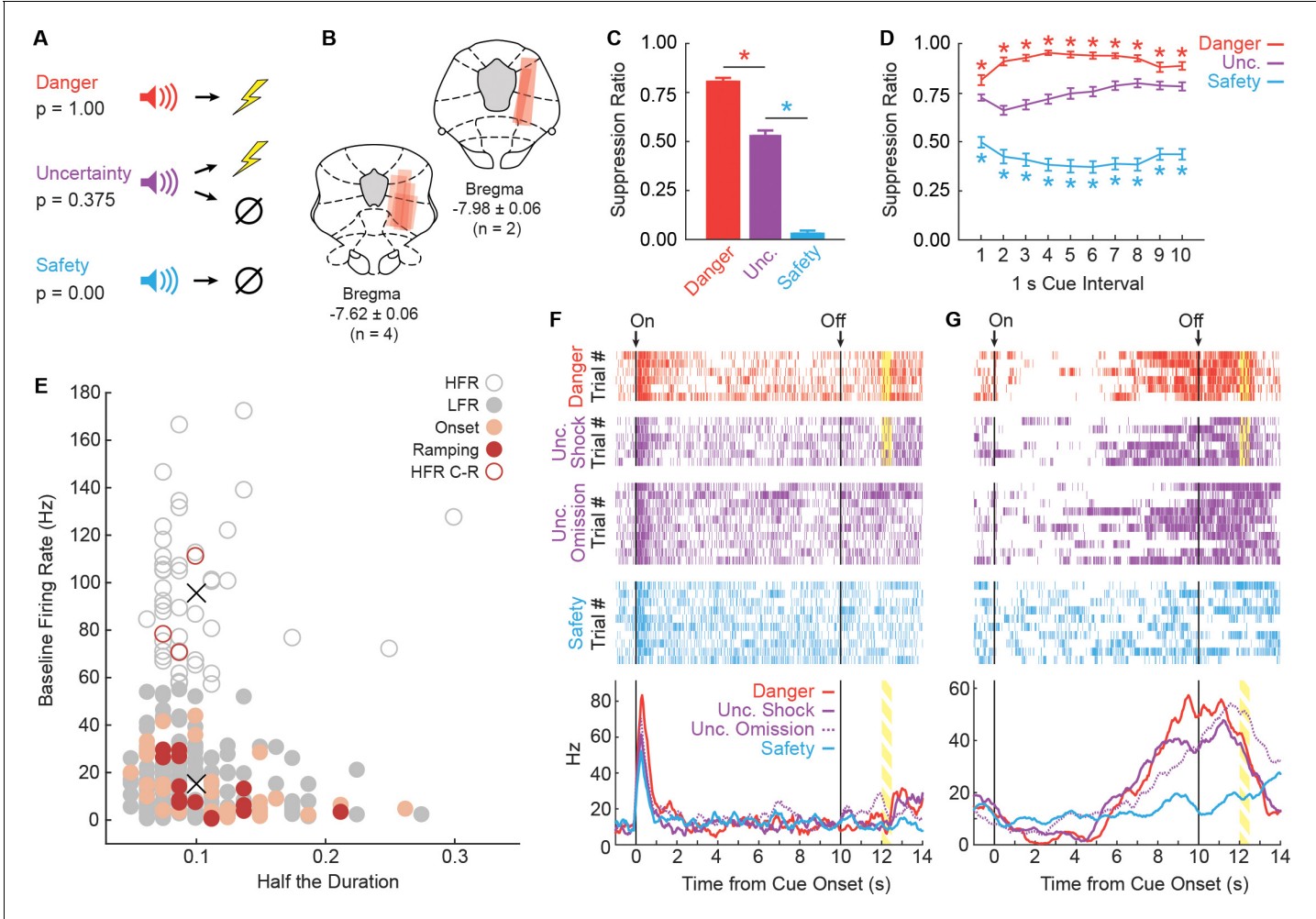

**Figure 1.** Fear discrimination, histology and vlPAG single-unit activity. (**A**) Pavlovian fear discrimination consisted of three cues, each predicting a unique probability of foot shock: danger, p=1.00 (red); uncertainty, p=0.375 (purple); and safety, p=0.00 (blue). (**B**) Microelectrode bundle placements for all rats (n = 6) and all neurons (n = 245) during recording sessions are represented by red bars. (**C**) Mean + SEM suppression ratio during the entire 10 s cue for danger, uncertainty, and safety trials is shown for all sessions in which single-units were recorded (n = 88). Discrimination was observed for each cue pair (danger vs. uncertainty, $t_{87}$ = 12.36, p=7.44×10$^{-21}$, red asterisk; uncertainty vs. safety, $t_{87}$ = 20.85, p=3.50×10$^{-35}$, blue asterisk). (**D**) Mean ± SEM suppression ratio during each 1 s interval of 10 s cue presentation for danger, uncertainty, and safety trials is shown (n = 88). Discrimination was observed during every interval for each cue pair (danger vs. uncertainty, all $t_{87}$ >3.00, all p<0.005 [Bonferroni correction for 10 tests], red asterisks; uncertainty vs. safety, all $t_{87}$ >7.00, all p<0.005, blue asterisks). (**E**) Scatterplot comparing half the duration of the waveform (x axis) to baseline firing rate (y axis) in all recorded neurons (n = 245). Clustering revealed two populations based on baseline firing rate (High Firing Rate (HFR), open circles; Low Firing Rate (LFR), solid gray circles). X symbols indicate cluster centroids. Onset neurons (n = 29, peach), Ramping neurons (n = 14, maroon) and HFR cue-responsive neurons (n = 3, maroon outline) are indicated. (**F and G**) Representative single-units from the (**F**) Onset population and (**G**) Ramping population. Cue onset (On) and offset (Off) indicated by vertical black lines. Shock delivery indicated by yellow bars. Trial-by-trial firing (top four raster plots), as well as mean firing (bottom, line graphs) are shown for each neuron. Each raster tick represents a spike and each row of spikes reflects one trial [danger (n = 6), uncertainty shock (n = 6), uncertainty omission (n = 10), safety (n = 10)]. The bottom row of spikes in each raster plot corresponds to the first cue trial, subsequent trials are above. Line graphs: Mean firing rate (Hz) across all trials for each cue was constructed using 100 ms bins and smoothed, cue boundaries and shock visualization maintained from raster plots above.

DOI: https://doi.org/10.7554/eLife.45013.003

The following figure supplements are available for figure 1:

**Figure supplement 1.** Fear discrimination measured by nose poke rate.
DOI: https://doi.org/10.7554/eLife.45013.004

**Figure supplement 2.** Imperfectly distributed nose poking inflates interval suppression ratios.
DOI: https://doi.org/10.7554/eLife.45013.005

**Figure supplement 3.** Firing and behavior for every recorded neuron.
DOI: https://doi.org/10.7554/eLife.45013.006

*Figure 1 continued on next page*

*Figure 1 continued*

**Figure supplement 4.** Waveform and firing characteristics of Onset neurons.
DOI: https://doi.org/10.7554/eLife.45013.007
**Figure supplement 5.** High firing cue-responsive units.
DOI: https://doi.org/10.7554/eLife.45013.008
**Figure supplement 6.** Waveform and firing characteristics of Ramping neurons.
DOI: https://doi.org/10.7554/eLife.45013.009

The electrode bundle was advanced ~40–80 μm between sessions to record from new single-units in subsequent sessions.

Across all 88 recording sessions, these six rats showed excellent discriminative fear: high to danger, intermediate to uncertainty, and low to safety (*Figure 1C*). ANOVA for suppression ratios for the total 10 s cue [factor: cue (danger vs. uncertainty vs. safety)] found a significant effect of cue ($F_{2,174}$ = 592.00, p=2.32×$10^{-78}$, $\eta_p^2$ = 0.87, observed power (op) = 1.00). Paired t-tests confirmed differing ratios for each cue (danger vs. uncertainty, $t_{87}$ = 12.36, p=7.44×$10^{-21}$; uncertainty vs. safety, $t_{87}$ = 20.85, p=3.50×$10^{-35}$). Visualizing nose poke rates to each cue revealed a discrimination pattern matching that for suppression ratio (*Figure 1—figure supplement 1*). While the foot shock probability associated with uncertainty was closer to safety: danger (p=1.00) >> uncertainty (p=0.375) > safety (p=0.00); the mean suppression ratio to uncertainty was closer to danger: danger (ratio = 0.80) > uncertainty (ratio = 0.53) >> safety (ratio = 0.03). The non-linear relationship between shock probability and behavior will be critical for regression analyses that seek to determine if single-unit firing is better captured by threat probability or fear output.

When the total cue period was divided into 10, 1 s intervals, discrimination was observed in each interval and its pattern showed subtle change over cue presentation (*Figure 1D*). ANOVA for suppression ratios [factors: cue (danger vs. uncertainty vs. safety) and interval (1-10)] found a significant effect of cue ($F_{2,174}$ = 489.40, p=3.59×$10^{-72}$, $\eta_p^2$ = 0.85, op = 1.00) and a cue x interval interaction ($F_{18,1566}$ = 6.21, p=5.90×$10^{-15}$, $\eta_p^2$ = 0.07, op = 1.00). Suppression ratios are artificially high when calculated in short intervals, making for a poor measure of *absolute* fear (*Figure 1—figure supplement 2*). However, this artificial skewing is observed equally to all cues, making suppression ratios in short intervals a valid, *relative* measure of fear.

## vlPAG neurons are responsive at cue onset or ramp over cue presentation

We recorded the activity of 245 neurons in six rats over 88 fear discrimination sessions. A previous study optogenetically identified vlPAG glutamate neurons (vGluT2+) as having low baseline firing rates, compared to GABA (gamma-aminobutyric acid) neurons (Gad1+) that exhibited high baseline firing rates (*Tovote et al., 2016*). We performed k-means clustering for all 245 neurons using baseline firing rate and the waveform characteristics: amplitude ratio and half the duration (*Roesch et al., 2007*). All neurons separated into one of two clusters purely on the basis of baseline firing rate (*Figure 1E*), with majority of neurons falling into the low firing rate (LFR) cluster (n = 199) and the remaining in the high firing rate (HFR) cluster (n = 46).

Independent of cluster membership, we determined the cue-responsiveness of each neuron (n = 245; cue firing of all neurons shown in *Figure 1—figure supplement 3*). Previous studies have identified a population of vlPAG neurons showing short-latency firing increases to auditory cues paired with foot shock (*Tovote et al., 2016*; *Watson et al., 2016*; *Ozawa et al., 2017*). We identified 29 neurons (obtained from 5/6 rats, ~12% of all neurons recorded) with phasic increases in firing to danger, uncertainty, *or* safety (t-test for firing rate, baseline [2 s prior to cue onset] vs. first 1 s cue interval, p<0.017, Bonferroni correction for three tests). All 29 neurons belonged to the LFR cluster and these neurons are referred to as the Onset population (single-unit example, *Figure 1F*; information for each Onset neuron's waveform, firing characteristics, subject and session can be found in *Figure 1—figure supplement 4*). Consistent with the most recent report (*Ozawa et al., 2017*), 17 neurons increased firing to at least one cue during the last 1 s cue interval (interval > baseline, p<0.017). Three neurons belonged to the high firing cluster and are likely a unique class of neurons (see *Figure 1—figure supplement 5* for a full analysis). The remaining 14 neurons (obtained from 4/6 rats, ~6% of all neurons recorded) belonged to the LFR cluster and are referred to as the Ramping

population (single-unit example, *Figure 1G*; information for each Ramping neuron's waveform, firing characteristics, subject and session can be found in *Figure 1—figure supplement 6*). All manuscript analyses were performed on these 29 Onset neurons and 14 Ramping neurons.

## vlPAG neurons show differential firing that is maximal to danger

Despite identifying neurons without regard for relative firing to the three cues, differential firing was observed in Onset neurons at the single-unit (*Figure 1F*) and population (*Figure 2A*) levels. Onset neurons (n = 29) sharply increased activity during the first 1 s cue interval, with greatest firing to danger, lesser firing to uncertainty, and least firing to safety (*Figure 2B*, Left). The differential firing pattern diminished as cue presentation proceeded and was absent by the last 1 s cue interval (*Figure 2B*, Right). The temporal firing pattern (onset → offset) was observed for all trials in the session (*Figure 2—figure supplement 1A–D*). ANOVA for normalized firing rate (Z-score transformation) for the 29 Onset neurons [data from *Figure 2A*; factors: cue (danger vs. uncertainty vs. safety) and bin (100 ms: 1 s prior to cue onset through 10 s cue)] revealed main effects of cue and bin (Fs > 9, ps<0.01, $\eta_p^2$ > 0.20, op > 0.95) but most critically, a cue x bin interaction ($F_{218,6104}$ = 1.94, p<0.01, $\eta_p^2$ = 0.06, op = 1.00). Consistent with the ANOVA interaction, Onset neurons showed significantly greater firing to danger compared to uncertainty in the first 1 s cue interval ($t_{28}$ = 4.54, p=9.70×10$^{-5}$). While numerically greater firing to uncertainty over safety failed to reach significance in the first 1 s interval ($t_{28}$ = 1.37, p=0.18), ANOVA restricted to uncertainty and safety (100 ms: 1 s prior to cue onset through first 5 s of the cue) revealed a significant cue x bin interaction ($F_{59,1652}$ = 1.50, p=0.01, $\eta_p^2$ = 0.051, op = 1.00). Differential firing was not observed to danger vs. uncertainty ($t_{28}$ = 1.69, p=0.10) or to uncertainty vs. safety ($t_{28}$ = 0.60, p=0.55) in the last 1 s cue interval.

Selective firing was observed in Ramping neurons at the single-unit (*Figure 1G*) and population (*Figure 2C*) levels. Ramping neurons (n = 14) did not increase firing to any cue during the first 1 s cue interval (*Figure 2D*, Left). Instead, activity ramped over cue presentation with greatest firing observed during the last 1 s cue interval (*Figure 2D*, Right). Ramping activity was most apparent to danger, intermediate to uncertainty, and absent to safety. The temporal firing pattern (onset → offset) was consistent across trials (*Figure 2—figure supplement 1E–H*). ANOVA for normalized firing rate for the 14 Ramping neurons [data from *Figure 2C*; factors: cue (danger vs. uncertainty vs. safety) and bin (100 ms: 1 s prior to cue onset through 10 s cue)] revealed main effects of cue and bin (Fs >60, ps <0.01, $\eta_p^2$ > 0.40, op = 1.00) and a cue x bin interaction ($F_{218,2834}$ = 4.33, p<0.01, $\eta_p^2$ = 0.24, op = 1.00). Illustrative of the ANOVA interaction, Ramping neurons showed no significant differences in firing to danger vs. uncertainty ($t_{13}$ = 0.62, p=0.55) or uncertainty vs. safety ($t_{13}$ = 0.24, p=0.81) in the first 1 s cue interval. However, differential firing to danger vs. uncertainty ($t_{13}$ = 3.17, p=7.41×10$^{-3}$), and uncertainty vs. safety ($t_{13}$ = 8.26, p=2.00×10$^{-6}$), was observed in the last 1 s cue interval.

Ramping activity to danger and uncertainty could be the product of the *time* at which activity began to increase, or the *rate* of increase. We performed a two-tailed t-test for population firing to danger vs. safety (*Figure 2E*, red line) and uncertainty vs. safety (*Figure 2E*, purple line) in a 1 s window, starting with cue onset. We slid the 1 s window across the 10 s cue in 100 ms increments, to reveal the time in which danger and uncertainty population firing departed from safety. We then calculated the rate of firing increase from the departure window to the last 1 s interval for danger and uncertainty. Differential firing was determined by the time of departure from safety, as opposed to the rate of increase. Ramping activity to danger emerged earlier (*Figure 2E*; 2.8 s following cue onset for p<0.05; 5.8 s for full Bonferroni correction, p<0.00025 [0.05/200]) than ramping activity to uncertainty (5.7 s following cue onset; 6.5 s for full Bonferroni correction). Change in firing rate did not differ between danger and uncertainty (*Figure 2E*, Inset).

It is possible that while Onset and Ramping neurons are distinct, activity of one population drives the other. We identified four Onset-Ramping pairs recorded in the same session. We then asked if trial-by-trial variations in firing over each 1 s cue interval were negatively correlated for any of the four pairs. We failed to uncover such a relationship (*Figure 3*), demonstrating that Onset and Ramping populations are distinct and independent.

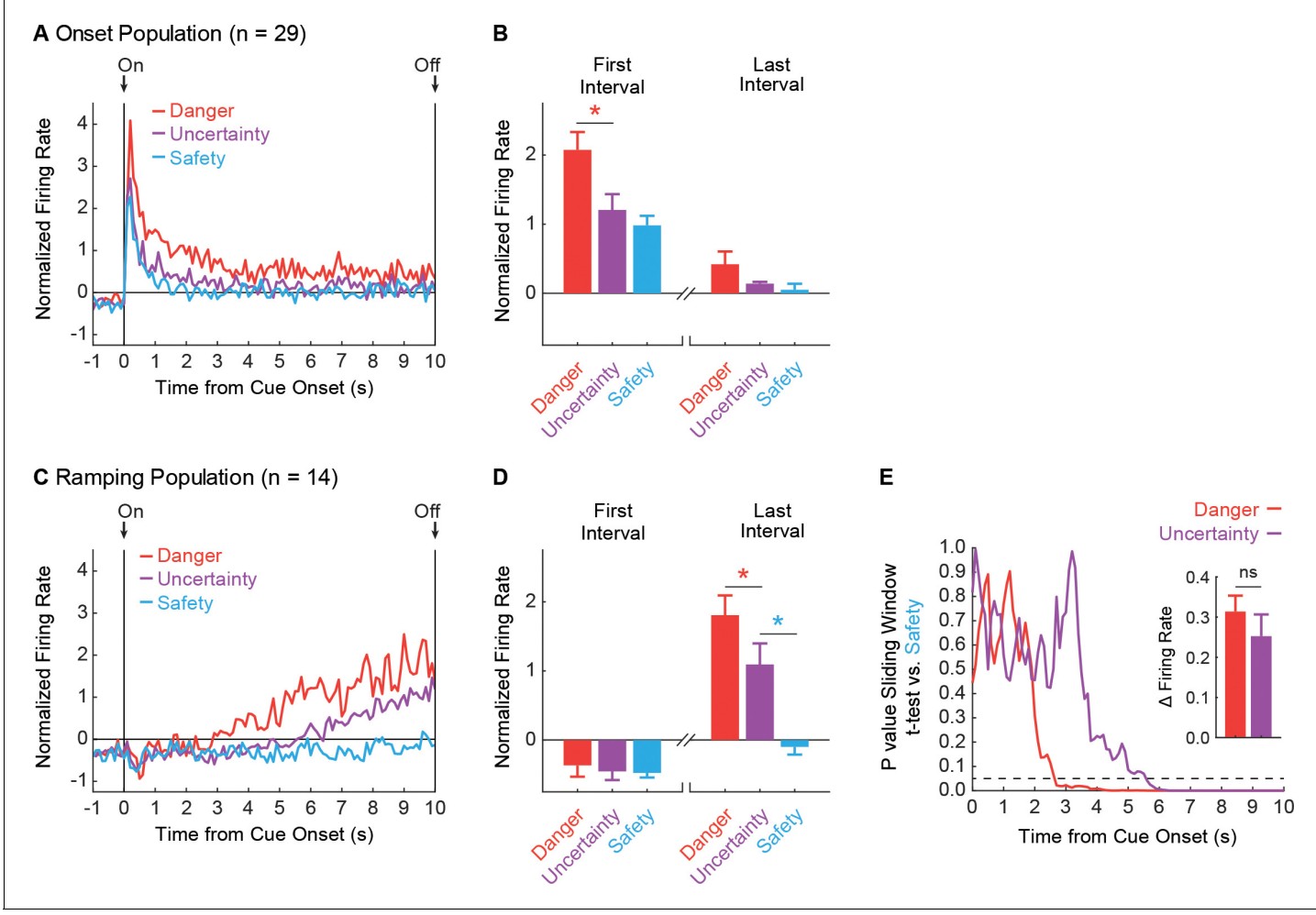

**Figure 2.** vlPAG neurons are responsive to cue onset or ramp over cue presentation. (**A**) Mean, Z-score normalized firing to danger (red), uncertainty (purple) and safety (blue) is shown for the 1 s pre-cue period and the 10 s cue period for the Onset population (n = 29). Cue onset (On) and offset (Off) are indicated by vertical black lines. (**B**) Mean + SEM, Z-score normalized firing during the first, 1 s cue interval (left) and the last, 1 s cue interval (right), is shown for each cue. Differential firing was observed for danger vs. uncertainty ($t_{28}$ = 4.54, p=9.70×10$^{-4}$, red asterisk) but not for uncertainty vs. safety ($t_{28}$ = 1.37, p=0.18), in the first interval. No differences were observed for danger vs. uncertainty ($t_{28}$ = 1.69, p=0.10) or uncertainty vs. safety, ($t_{28}$ = 0.60, p=0.55) in the last interval. (**C**) Normalized firing for the Ramping population (n = 14) plotted as in A. (**D**) First and last interval firing for the Ramping population (n = 14) plotted as in B. Differential firing was not observed for danger vs. uncertainty ($t_{13}$ = 0.62, p=0.55) or uncertainty vs. safety ($t_{13}$ = 0.24, p=0.82), in the first interval. By contrast, differential firing was observed for danger vs. uncertainty ($t_{13}$ = 3.17, p=7.41×10$^{-3}$, red asterisk) and uncertainty vs. safety ($t_{13}$ = 8.26, p=2.00×10$^{-6}$, blue asterisk), in the last interval. (**E**) A t-test comparing danger (red) and uncertainty (purple) population firing to safety in a 1 s window was slid across the 10 s cue in 100 ms increments. P value of t-test reported on y axis. Dotted line indicates p=0.05. Inset: Mean + SEM change in firing rate from the first window of activity departed from safety to the last interval, is shown for danger (red) and uncertainty (purple). ns = no significance of a paired t-test.

DOI: https://doi.org/10.7554/eLife.45013.010

The following figure supplements are available for figure 2:

**Figure supplement 1.** Trial by trial firing for Onset and Ramping populations.
DOI: https://doi.org/10.7554/eLife.45013.011

**Figure supplement 2.** Nose poke cessation is insufficient to drive activity of Onset and Ramping neurons.
DOI: https://doi.org/10.7554/eLife.45013.012

## Population biases are evident in vlPAG single-units

To demonstrate that Onset population activity was the result of a consistent bias across neurons, we directly compared single-unit firing to cue pairs. Danger and uncertainty firing were correlated, and single-units were biased towards greater firing to danger (*Figure 4A*). Uncertainty and safety firing

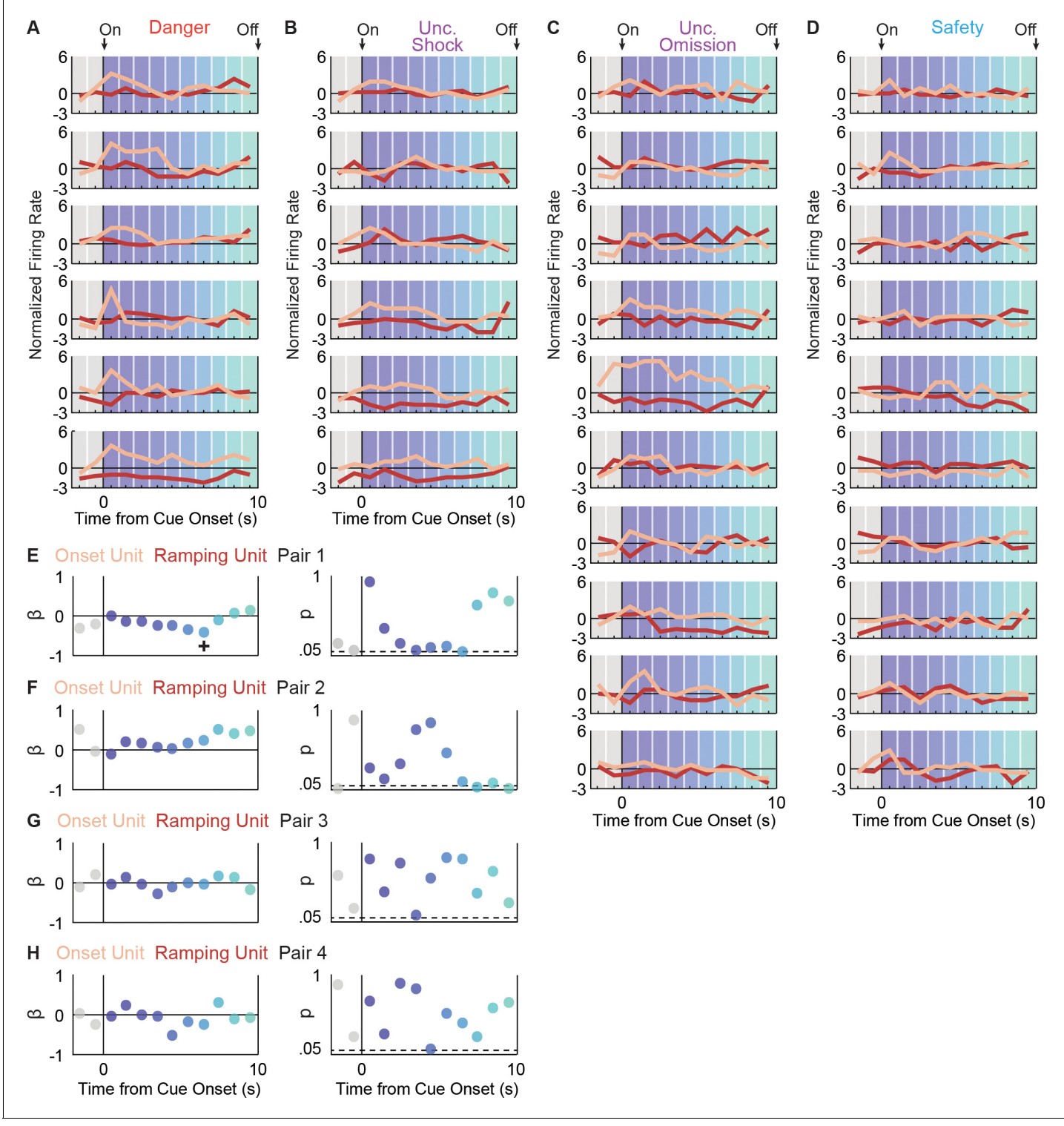

**Figure 3.** Onset and Ramping single-unit activity are independent. Activity was simultaneously recorded for a Ramping (maroon) and Onset (peach) unit in four separate sessions (four pairs). For Pair 1, activity is shown for each trial type: (**A**) danger, (**B**) uncertainty shock, (**C**) uncertainty omission, and (**D**) safety. Single-unit regression was performed for each interval. The primary regression output was a beta coefficient (β) and a p value (p) for: (**E**) pair 1, (**F**) pair 2, (**G**) pair three and (**H**) pair 4. A significant (p<0.05), negative β was only found for one interval, for one neuron pair (pair 1, interval indicated by +).

DOI: https://doi.org/10.7554/eLife.45013.013

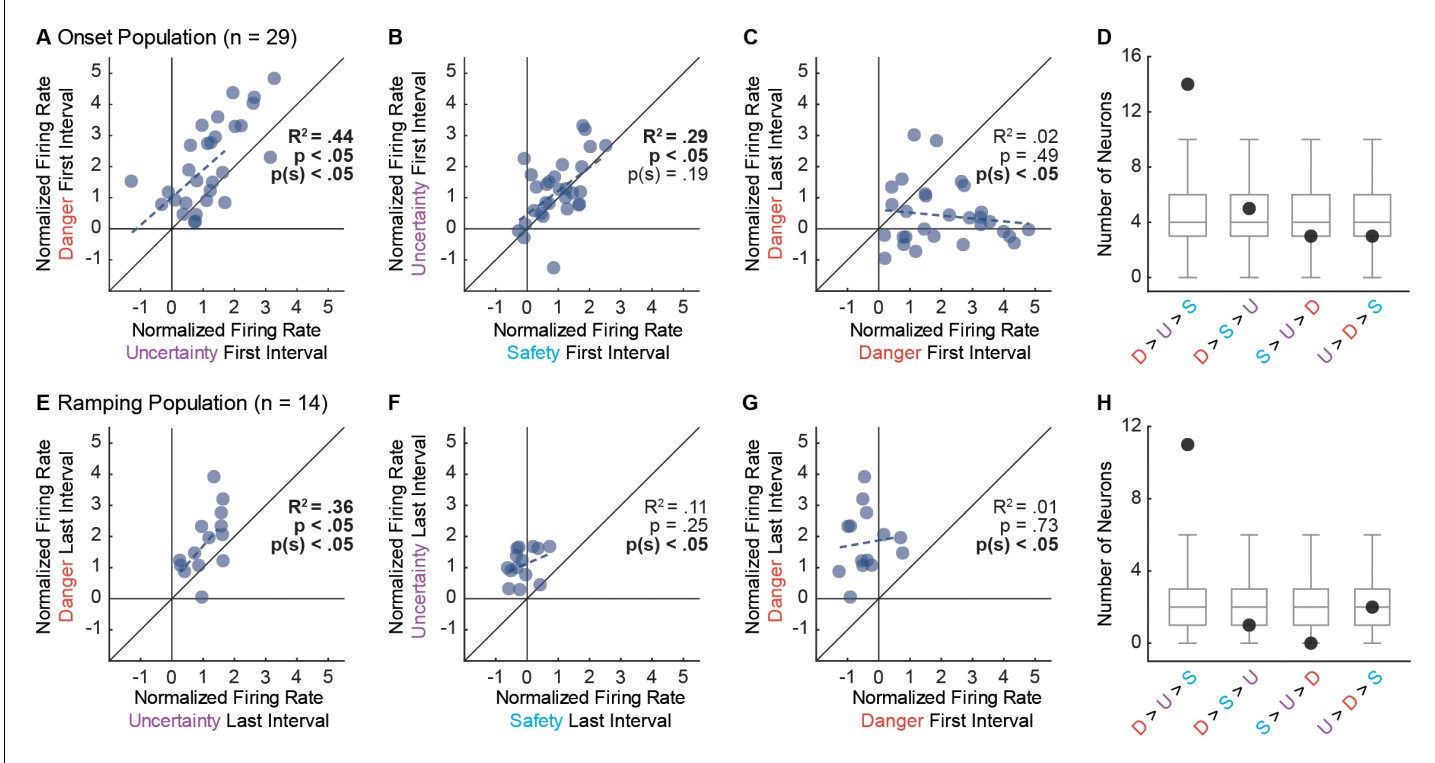

**Figure 4.** Single-unit biases in Onset and Ramping populations. (**A**) Normalized firing to uncertainty (purple) vs. danger (red) during the first, 1 s cue interval is plotted for all Onset neurons (n = 29). Trendline, the square of the Pearson correlation coefficient ($R^2$) with associated p value (p), and sign test p value p(s) are shown for each plot. (**B**) Normalized firing to safety (blue) vs. uncertainty (purple) during the first, 1 s cue interval is plotted for Onset neurons (n = 29). (**C**) Normalized firing to danger in the first, 1 s cue interval vs. the last, 1 s cue interval is plotted for Onset neurons (n = 29). (**D**) Number of observed neurons (closed circle) vs shuffled distribution (median, 25th percentile, 75th percentile, lowest/highest non-outliers) shown for four most common firing patterns: danger > uncertainty > safety, danger > safety > uncertainty, safety > uncertainty > danger, and uncertainty > danger > safety. (**E**) Normalized firing to uncertainty (purple) vs. danger (red) during the last, 1 s cue interval is plotted for Ramping neurons (n = 14). (**F**) Normalized firing to safety (blue) vs. uncertainty (purple) during the last, 1 s cue interval is plotted for Ramping neurons (n = 14). (**G**) Normalized firing to danger in the first, 1 s cue interval vs. the last, 1 s cue interval is plotted for Ramping neurons (n = 14). (**H**) Number of observed neurons vs shuffled distribution reported as in D.

DOI: https://doi.org/10.7554/eLife.45013.014

were also correlated; however, the single-unit bias towards greater firing to uncertainty was not significant (*Figure 4B*). Underscoring their specificity to cue onset, single-units were biased towards greater firing to danger in the first 1 s cue interval compared to the last interval, and there was no correlation between firing in the two epochs (*Figure 4C*). Examining relative cue firing for each Onset neuron in the first 1 s interval revealed the most common pattern to be: danger > uncertainty > safety (n = 14). This was the only pattern to contain more units than would be expected than chance (*Figure 4D*).

We performed the same analysis for the Ramping population, only for the last 1 s interval. Ramping neurons showed a selective firing pattern. A significant correlation between firing to danger and uncertainty was observed, along with a single-unit bias towards greater firing to danger (*Figure 4E*). Only now, there was no correlation between uncertainty and safety firing, but a consistent bias towards greater uncertainty firing (*Figure 4F*). Ramping single-units were biased towards danger activity in the last 1 s cue interval, and there was no correlation between firing in the two epochs (*Figure 4G*). The most common firing pattern in the last 1 s interval was: danger > uncertainty > safety (n = 11). This was the only pattern to contain more units than expected by chance (*Figure 4H*).

VlPAG activity is greatest to danger, the cue most strongly suppressing rewarded nose poking. It is therefore possible that Onset and Ramping neurons are simply responsive to nose poke cessation.

To examine this possibility, we identified naturally occurring periods of nose poke cessation in inter-trial intervals, when no cues were presented. This analysis found no meaningful changes in Onset or Ramping activity during periods of nose poke cessation (*Figure 2—figure supplement 2*), demonstrating activity patterns are specific to cue-induced suppression of nose poking.

At first glance, the firing patterns of Onset and Ramping neurons appear to support the prevailing hypothesis that vlPAG neurons signal fear output. Differential fear (*Figure 1C*) and differential firing (*Figure 2B,D*) show the same general pattern: danger > uncertainty > safety. However, closer inspection reveals that relative differences in fear do not match relative differences in firing. Rats showed robust discrimination between uncertainty and safety, regardless of the temporal resolution with which fear was measured (*Figure 1C,D*). Yet, robust differential firing to uncertainty and safety was modest in the Onset population (*Figure 2B*, left; *Figure 4B*). The Ramping population showed stronger differential firing between uncertainty and safety (*Figure 2D*, right; *Figure 4F*), but this pattern did not emerge in until the end of the cue. Fear discrimination was reliably detected in the first 1 s interval (*Figure 1D*), indicating that Ramping neurons cannot organize fear output early in cue presentation. While inconsistencies between fear output and neural activity are evident for the Onset and Ramping populations, the analyses conducted so far cannot conclusively test the relative contribution of threat probability and fear output to vlPAG single-unit activity.

## Onset neurons signal threat probability

To formally test the degree to which vlPAG activity is captured by fear output and threat probability, we used simultaneous linear regression for single-unit firing (*Figure 5*; see Materials and methods for example regression input). For each single-unit, we calculated the normalized firing rate for each trial (32 total: six danger, six uncertainty shock, 10 uncertainty omission, and 10 safety), in 1 s bins over the 10 s cue. For each trial, we calculated fear on two time scales: total fear (suppression ratio for the entire 10 s cue) and interval fear (suppression ratio for the specific 1 s interval). The corresponding shock probability was assigned for each trial (danger = 1.00, uncertainty = 0.375 and safety = 0.00). Total fear, interval fear and threat probability were used as regressors to explain trial-by-trial variance in single-unit firing. Statistical output was a beta coefficient quantifying the strength (|>0| = stronger) and direction (>0 = positive) of the predictive relationship between each regressor and single-unit firing. Beta coefficients for single-units comprising the Onset and Ramping populations were subjected to ANOVA with regressor (total fear vs. interval fear vs. probability) and interval (1 s intervals for 10 s cue) as factors. This allowed us to determine the relative contribution of total fear, interval fear and probability to single-unit firing over the course of cue presentation.

The results of primary interest for the Onset population came from the first 1 s cue interval, when activity was highest and differential firing was observed. Linear regression unequivocally revealed that Onset single-unit activity was captured by threat probability (*Figure 5A*). The beta coefficient for the probability regressor was positive and significant, exceeding the beta coefficient for either measure of fear output – neither of which differed from zero. The population bias was observed across Onset neurons. Single-unit beta coefficients were positively biased for threat probability, but not for either measure of fear output (*Figure 5B,C*).

Examining the entirety of cue presentation, threat probability signaling was highest in the first interval, persisted several more seconds and diminished (*Figure 5D*). Total fear or interval fear did not account for variance in single-unit firing at any interval. Consistent with this description, ANOVA for beta coefficient with factors of regressor and interval (10 total) revealed a main effect of regressor ($F_{2,56}$ = 7.16, p<0.01, $\eta_p^2$ = 0.20, op = 0.92) and a regressor x interval interaction ($F_{18,504}$ = 2.38, p<0.01, $\eta_p^2$ = 0.08, op = 0.99). Identical results were obtained when probability was compared to total fear or interval fear separately. In fact, significance for fear output could only be found if total fear was the *only* regressor used in the analysis – producing a result very similar to that of a previous study (*Watson et al., 2016*). Even then, the predictive relationship was weaker than that of probability (*Figure 5—figure supplement 1, A–E*).

The threat probability regressor in the above analyses utilized the actual shock probability assigned to each cue. Of course, the subjects had no *a priori* knowledge of shock probability assignments. It is then possible that vlPAG activity is 'tuned' to an alternative shock probability. To examine this, we performed single-unit linear regression for normalized firing in the first 1 s cue interval maintaining the probabilities for danger (1.00) and safety (0.00), but incrementing the probability assigned to uncertainty from 0 to 1 in 0.125 steps (0.000, 0.125, 0.250, 0.375, 0.500, 0.625, 0.750,

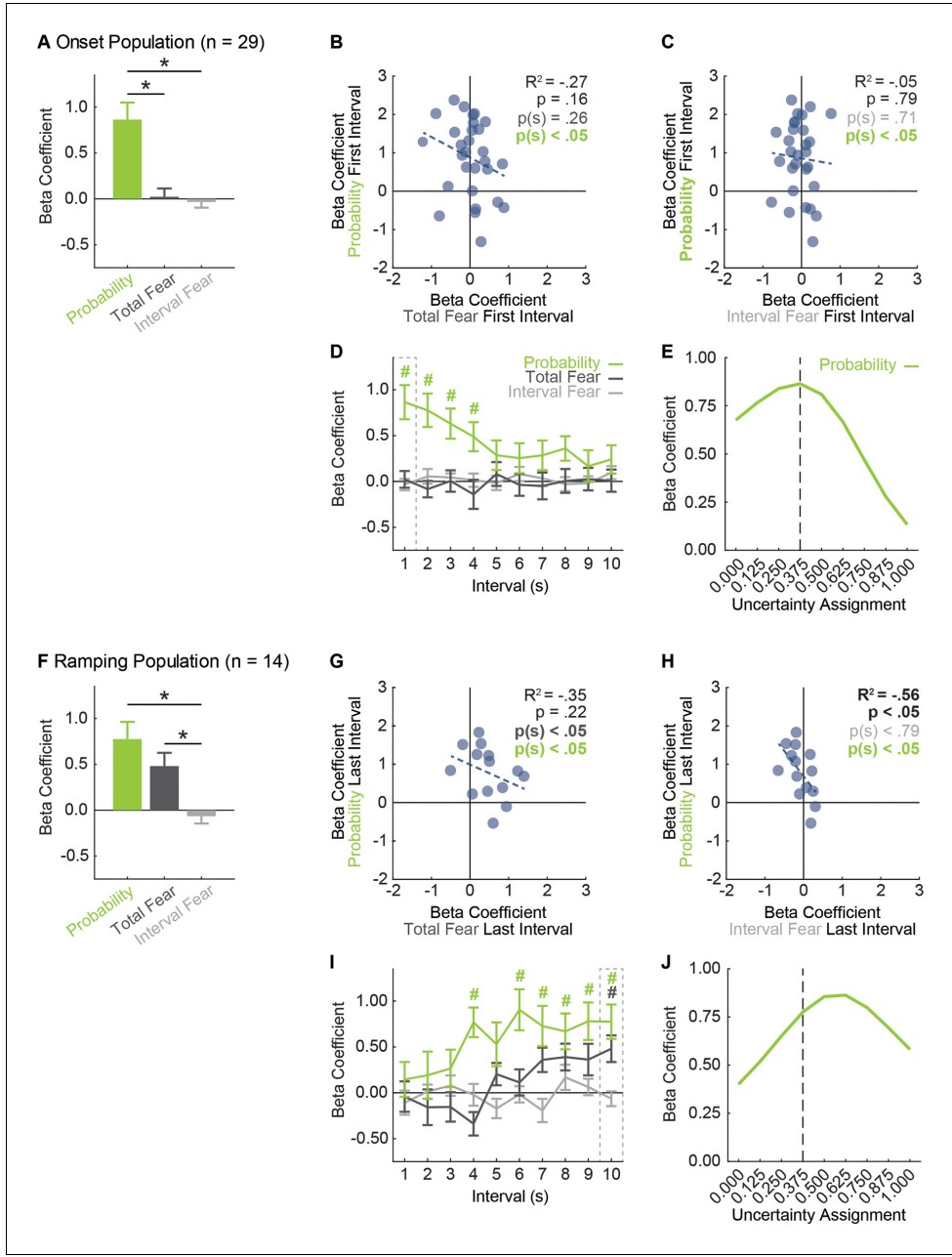

**Figure 5.** vlPAG neurons prioritize threat probability over fear output. (**A**) Mean + SEM beta coefficient is shown for each regressor, during the first, 1 s cue interval, for the Onset population (n = 29): probability (green), total fear (dark gray) and interval fear (light gray). Beta coefficient for probability was significantly greater than either measure of fear (probability vs. total fear, $t_{28}$ = 3.75, p=8.01×10$^{-4}$; probability vs. interval fear, $t_{28}$ = 4.57, p=8.90×10$^{-5}$). *paired samples t-test, p<0.05. (**B**) Beta coefficient for total fear (dark gray) vs. probability (green) during the first, 1 s cue interval is plotted for all Onset neurons (n = 29). Trendline, the square of the Pearson correlation coefficient ($R^2$) with associated p value, and sign test p value comparing each regressor to zero is shown. (**C**) Beta coefficient for interval fear (light gray) vs. probability (green) during the first, 1 s cue interval is plotted for all Onset neurons (n = 29). Sign test result for probability is identical to B. (**D**) Mean ± SEM beta coefficient is shown for each regressor, during each 1 s cue interval, for the Onset population (n = 29). Dash outlined box indicates interval analyzed in (**A**). #single-sample t-test comparison to zero, p<0.005 (Bonferroni correction for 10 tests), color indicates regressor compared to zero. (**E**) Mean beta coefficient for probability is shown for the each of nine uncertainty assignments for the Onset population (n = 29). Dashed line indicates the actual foot shock probability associated with uncertainty (0.375). (**F**) Mean + SEM beta coefficient is shown for each regressor, during the last, 1 s cue interval, for the Ramping population (n = 14). Beta coefficients for probability

*Figure 5 continued on next page*

*Figure 5 continued*

and total fear were significantly greater than interval fear (probability vs. interval fear, $t_{13}$ = 3.61, p=3.20×10$^{-3}$; total fear vs. interval fear, $t_{13}$ = 4.68, p=4.34×10$^{-4}$), but did not differ from one another ($t_{13}$ = 1.11, p=0.29). *paired samples t-test, p<0.05. (G) Beta coefficient for total fear (dark gray) vs. probability (green) during the last, 1 s cue interval is plotted for Ramping neurons (n = 14). (H) Beta coefficient for interval fear (light gray) vs. probability (green) during the last, 1 s cue interval is plotted for Ramping neurons (n = 14). Sign test result for probability is identical to F. (I) Mean ± SEM beta coefficient is shown for each regressor, during each 1 s cue interval, for the Ramping population (n = 14). Dash outlined box indicates interval analyzed in (E). #single-sample t-test comparison to zero, p<0.005 (Bonferroni correction for 10 tests), color indicates regressor compared to zero. (J) Mean beta coefficient for probability is shown for the each of nine uncertainty assignments for the Ramping population (n = 14). Dashed line indicates the actual foot shock probability associated with uncertainty (0.375).

DOI: https://doi.org/10.7554/eLife.45013.015

The following figure supplements are available for figure 5:

**Figure supplement 1.** Onset and Ramping single-units prioritize threat probability irrespective of regressor combination.

DOI: https://doi.org/10.7554/eLife.45013.016

**Figure supplement 2.** Onset single-unit activity is better captured by threat probability than a binary output.

DOI: https://doi.org/10.7554/eLife.45013.017

---

0.875, and 1.000). The mean beta coefficient for each of the nine increments is plotted as a threat-tuning curve for the Onset population (*Figure 5E*). The beta coefficient resulting from regression using the actual shock probability (uncertainty = 0.375), was the 'peak' of the tuning curve. The probabilities with the next highest beta coefficients were those flanking 0.375. Beta coefficients dropped off rapidly as the uncertainty assignment moved to the extremes.

This result is particularly revealing for the analysis in which the uncertainty assignment was 0.000 (first data point on the curve *Figure 5E*). Onset neurons showed high firing to danger but lower and more similar firing to uncertainty and safety, leaving open the possibility that Onset neurons signal a more binary output (danger = 1.000) > (uncertainty and safety = 0.000). However, the actual uncertainty assignment (0.375) captured single-unit activity better than the binary assignment (0.000) in the first 1 s interval and across the remainder of cue presentation (*Figure 5—figure supplement 2*).

## Ramping neurons prioritize threat probability over fear output

Linear regression for the Ramping population in the last 1 s cue interval revealed that single-unit activity was captured by a mixture of threat probability and total fear output (*Figure 5F*). Ramping single-units were biased towards positive beta coefficients for probability and total fear (*Figure 5G*), but there was no correlation between these regressors. Ramping single-units were not biased towards positive beta coefficients for interval fear (*Figure 5H*), but signaling of probability and interval fear were negatively correlated. Linear regression for all ten intervals revealed that threat probability signaling was prioritized over total fear and interval fear (*Figure 5I*). ANOVA for beta coefficients with factors of regressor and interval revealed a main effect of regressor ($F_{2,26}$ = 8.96, p<0.01, $\eta_p^2$ = 0.41, op = 0.96) and regressor x interval interaction ($F_{18,234}$ = 2.16, p<0.01, $\eta_p^2$ = 0.14, op = 0.99). Mean ± SEM beta coefficients for each regressor over all 10 intervals were: threat probability: 0.58 ± 0.14, total fear: 0.12 ± 0.10 and interval fear: −0.03 ± 0.05. Only the beta coefficient for probability differed significantly from zero ($t_{13}$ = 4.21, p=1.02×10$^{-3}$) and was significantly greater than either total fear ($t_{13}$ = 2.32, p=0.04) or interval fear ($t_{13}$ = 4.32, p=8.28×10$^{-4}$). This pattern of results held when probability was separately compared to total fear and interval fear, and when each regressor was considered in isolation (*Figure 5—figure supplement 1, F–J*).

If Ramping neurons contain information about threat probability, as well as fear output, the tuning curve for ramping neurons ought to be shifted right of 0.375. This is because the relative weighting of uncertainty for threat probability (danger >> uncertainty > safety) and average fear output (danger > uncertainty >> safety) differ. We constructed a population threat-tuning curve for normalized firing in the last 1 s interval (*Figure 5J*, as in *Figure 5E*). Tuning was shifted right of the actual probability, with a 'peak' at 0.625. This is consistent with mixed signaling of fear output and threat probability by Ramping neurons, rather than a pure threat probability signal.

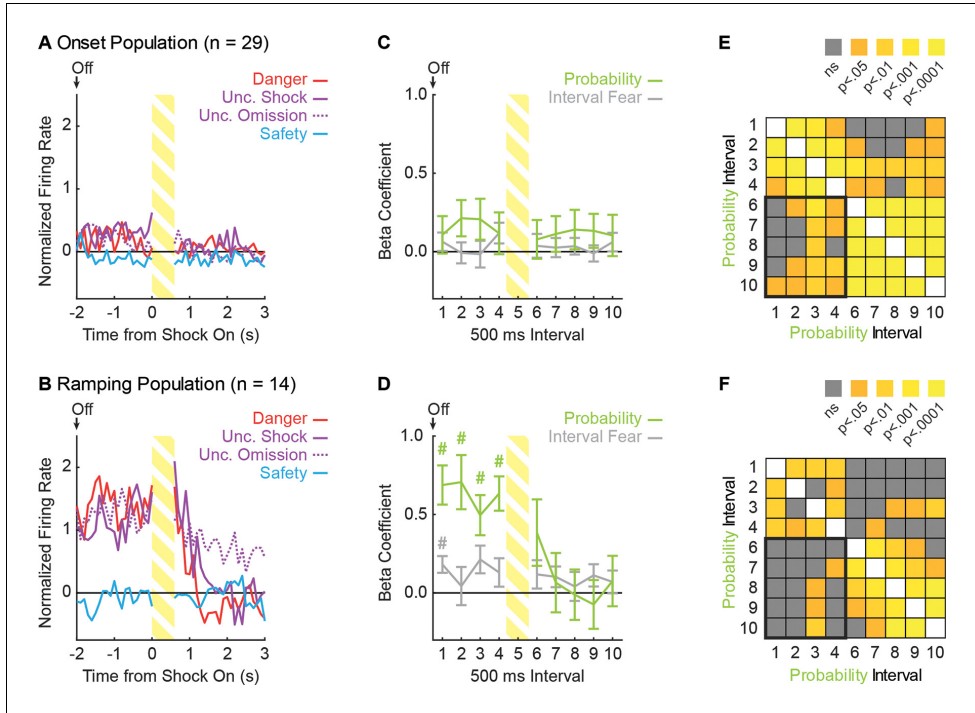

**Figure 6.** Ramping neurons signal threat probability until shock delivery. (**A**) Mean normalized firing to danger (red), uncertainty shock (purple, solid line), uncertainty omission (purple, dashed line) and safety (blue) is shown for the 5 s post-cue period for the Onset population (n = 29). Cue offset (Off) is indicated by the arrow, shock indicated by striped yellow bar. (**B**) Mean normalized firing is shown for the Ramping population (n = 14), as in A. (**C**) Mean ± SEM beta coefficient is shown for each regressor, over the 5 s post-cue interval, for the Onset population (n = 29): probability (green) and interval fear (light gray). (**D**) Mean ± SEM beta coefficient is shown for each regressor, over the 5 s post-cue interval, for the Ramping population (n = 14). #single-sample t-test comparison to zero, p<0.005 (Bonferroni correction for 10 tests), and color indicates regressor compared to zero. (**E**) A correlation matrix was constructed for the 29 Onset neurons using the probability beta coefficient from the four delay and five post-shock intervals. The p value of the Pearson correlation coefficient is plotted for each interval compared (legend shown next to figure). (**F**) A correlation matrix was constructed for the 14 Ramping neurons as in E.

DOI: https://doi.org/10.7554/eLife.45013.018

## Ramping neurons signal threat probability through delay until shock receipt

Onset neurons are tuned to initial cue presentation, providing a rapid estimate of threat probability. By contrast, Ramping neurons are initially unresponsive, but gradually increase activity over cue presentation. Ramping neurons may continue to signal threat probability through the delay period, up until foot shock receipt. We first examined population activity during the five seconds following cue offset (2 s delay, 0.5 s shock and 2.5 s post-shock). Activity during the 500 ms shock period was not analyzed because it may have been contaminated by electrical artifacts. Onset neurons showed negligible delay activity and little to no post-shock activity (*Figure 6A*). Ramping neurons continued firing to danger and uncertainty throughout the delay period, but this firing diminished shortly after shock presentation (*Figure 6B*). ANOVA for normalized firing rate [factors: trial type (danger vs. uncertainty shock vs. uncertainty omission vs. safety) and bin (100 ms: 5 s post cue)] revealed no main effects or interactions for Onset neurons ($F_s$ <2.3, ps >0.09). By contrast, identical ANOVA revealed main effects of trial and bin (Fs >3, ps <0.01), as well as a trial x bin interaction for Ramping neurons ($F_{132,1716}$ = 3.14, p<0.01, $\eta_p^2$ = 0.19, op >0.99). The Onset and Ramping firing patterns differed from one another. ANOVA with neuron type as a factor (Onset vs. Ramping) found significant interactions for trial type x neuron type, bin x neuron type, and trial type x bin x neuron type (Fs >2.50, ps <0.01, $\eta_p^2$ > 0.19, op >0.94).

Observed differences in neural activity suggest that Ramping neurons maintain threat probability signaling throughout the delay period and this signal abruptly decreases following foot shock presentation. Single-unit regression was used to determine whether trial-by-trial, post-cue firing was best described by interval fear or probability (total fear sampled only the prior 10 s cue period and was omitted). Regression was performed in 500 ms intervals for the 5 s post-cue period minus the shock interval (nine total intervals). Onset neurons did not signal interval fear or probability at any time following cue offset (*Figure 6C*). ANOVA revealed no main effect of regressor ($F_{1,28}$ = 1.02, p=0.32, $\eta_p^2$ = 0.04, op = 0.16) or regressor x interval interaction ($F_{8,224}$ = 0.53, p=0.83, $\eta_p^2$ = 0.02, op = 0.24). Illustrative of the lack of information contained in Onset neurons, post hoc comparisons found that no beta coefficient, for *any* regressor differed from zero. By contrast, Ramping neurons signaled probability throughout the delay period (*Figure 6D*, first four intervals), which diminished following shock delivery. In support, ANOVA revealed a significant regressor x interval interaction ($F_{8,104}$ = 5.20, p<0.01, $\eta_p^2$ = 0.29, op = 0.99).

Threat probability signaling prior to shock delivery was abruptly halted following shock delivery in Ramping neurons, but not Onset neurons. To reveal the degree to which this occurred, we constructed correlation matrices using the probability beta coefficient from each of the nine relevant intervals (4 delay and five post-shock). Of greatest interest were the twenty comparisons between the probability beta coefficient for the four delay and five post-shock intervals (*Figure 6E & F*; bottom-left quadrant). Significant between-interval correlations for the probability beta coefficient were observed for 13/20 comparisons in Onset neurons (*Figure 6E*), indicating that threat probability signaled during the delay tended to persist following foot shock. By contrast, significant correlations were found for only 4/20 comparisons in Ramping neurons (*Figure 6F*); and the proportion of intervals showing a significant correlation differed for the Onset and Ramping neurons ($\chi^2$ = 8.08, p=4.5×10$^{-3}$). Bonferroni correction (0.05/5 = 0.01, five tests per interval) found significant correlations for 6/20 Onset comparisons, 1/20 Ramping comparisons, and these proportions significantly differed ($\chi^2$ = 4.22, p=0.039).

## Discussion

We recorded vlPAG single-unit activity while rats discriminated between danger, uncertainty and safety. Consistent with previous reports (*Tovote et al., 2016*; *Watson et al., 2016*; *Ozawa et al., 2017*), we found a population of Onset neurons with short-latency excitation to danger. Consistent with the most recent report (*Ozawa et al., 2017*), we found a Ramping population that increased activity over danger presentation. Onset activity reflected an estimate of threat probability, invariant of fear output. Ramping activity reflected threat probability and fear output, though probability emerged earlier and was stronger overall. While vlPAG signals for fear output could potentially emerge at the ensemble level (*Jones et al., 2007*; *Zhou et al., 2018*), these multi-unit codes would be composed of single-units primarily signaling threat probability. Activity reflecting fear output may be found in other vlPAG populations, such as neurons showing inhibition of firing to cues (*Tovote et al., 2015*), or in non-cue-responsive single-units (*Insanally et al., 2019*). Yet, this would still mean that signals for threat probability and fear output co-exist in the vlPAG.

It is important to note that these results are correlative and that Onset neuron activity may not play a causal role in fear output. Previous work has found that short-latency, excitatory responses to danger are largely observed in vlPAG VGlut2 +neurons, and that excitation of this population is sufficient to produce freezing (*Tovote et al., 2016*). Though all Onset neurons fell into the low firing cluster, we cannot conclude these were VGlut2 +neurons. Moreover, inhibition of vlPAG GABA neurons also promotes freezing (*Tovote et al., 2016*). These neurons have comparatively higher baseline firing rates and respond to danger cues through inhibition of neural activity. A causal, vlPAG signal for fear output may be observed in a GABAergic/cue-inhibited population, in concert with or independent of a glutamatergic/cue-excited population. At the same time, it is clear how the observed Onset signal could play a causal role in fear output. Blocking vlPAG Onset activity to danger and uncertainty would equate neural activity to that for safety, removing the threat impetus for suppression of reward seeking, freezing and related defensive behaviors.

Before further discussing implications, we consider some alternative accounts for the observed firing patterns. Perhaps vlPAG neurons signaling fear output are anatomically distinct from those recorded here. We intentionally recorded from caudal vlPAG, the subregion preferentially activated

by fearful contexts in rats (*Carrive et al., 1997*). VlPAG manipulations that disrupt fear-related behaviors typically include this more caudal region (*De Oca et al., 1998*), and high-resolution functional magnetic resonance imaging reveals caudal vlPAG activation specific to aversive stimuli in humans (*Satpute et al., 2013*). We observed threat probability signaling in all recording locations (bregma −7.62 → −7.98). The vlPAG stretches ~0.7 mm beyond our most caudal recording site. It is therefore possible that neurons signaling fear output are restricted to the extreme caudal vlPAG.

Maybe the vlPAG signals fear output, but we did not measure the relevant output. Previous studies have failed to find robust relationships between vlPAG activity and cued freezing. Here we used conditioned suppression of rewarded nose poking to provide an objective measure of fear on two timescales and to perhaps better capture vlPAG activity. This measure of fear did not capture Onset neuron firing, and only partially captured Ramping neuron firing at the end of cue presentation. Further, Onset and Ramping activity were not merely driven by nose poke cessation or withdrawal from the port.

If not freezing, conditioned suppression, or nose poke cessation then perhaps another measure of fear? Danger cues elicit active fear responses: escape-like behaviors such as darting (*Greiner et al., 2018*; *Gruene et al., 2015*). However, darting is prevalent in females, but less so in males. Further, the males used in this study had extensive experience with fear discrimination, and at no point was escape from the foot shock possible. Danger cues increase arterial blood pressure and reduce heart rate, however, neither of these abilities has been linked to vlPAG function (*Helmstetter and Tershner, 1994*; *LeDoux et al., 1988*; *Wilson and Kapp, 1994*). Danger cues also enhance startle, perhaps more in line with Onset population firing. Yet, dorsal, rather than ventrolateral, PAG subregions have been implicated in startle behavior (*Walker et al., 1997*; *Zhao et al., 2009*). Of course, many other fear behaviors are possible: piloerection, hyperventilation, changes in body temperature, vocalization, etc. (*Kim et al., 2010*; *Iwata and LeDoux, 1988*; *Gallego et al., 2001*; *Vianna and Carrive, 2005*). Problematic is that most fear behaviors are initiated at cue onset and maintained until the aversive event occurs. We did not observe a substantial population of neurons with these temporal firing characteristics, making the vlPAG a poor candidate for sustained fear output. Above all, any potential behavior signaled by Onset neurons must closely match shock probability, confounding this behavior signal with probability itself.

The present findings are best understood through comparison to the account of vlPAG function outlined in the predatory imminence continuum (PIC), a highly influential theory of defensive behavior (*Fanselow and Lester, 1988*). Organizing features of the PIC are time and degree of threat. As predation becomes more imminent (pre-encounter → post-encounter → circa-strike), the form and intensity of defensive behaviors change. Cued fear is argued to capture post-encounter defenses: immobility elicited when predators are nearby. In the neural instantiation of PIC, the amygdala integrates information about environmental stimuli (auditory cues here), nociceptive information (foot shock) and time to produce a signal for degree of threat (*Fanselow and Lester, 1988*). This amygdala-derived signal is relayed to the vlPAG to organize fear output (*Fanselow, 1991*; *Fanselow, 1994*). Implicit in the PIC model, is that the vlPAG does not contain information about degree of threat – only the resultant fear output. Yet, we find that single vlPAG neurons contain detailed information about time and degree of threat.

These results require more careful consideration of the role of the vlPAG in the fear circuit and the PIC. Rather than signaling fear output, vlPAG Onset neurons signal threat probability. This information *could* be used to organize a variety of fear responses, but these neurons do not intrinsically signal fear output. For example, vlPAG projections to the central amygdala (CeA) and rostral ventromedial medulla (RVM) could inform fear output via freezing (*Vianna et al., 2008*), while projections to midline/intralaminar thalamus could rapidly relay threat probability estimates to a larger fear network (basolateral amygdala, prelimibic cortex, infralimbic cortex, insular cortex, etc.) (*Vertes et al., 2015*; *Krout and Loewy, 2000*; *Buchanan and Thompson, 1994*; *Sengupta and McNally, 2014*), promoting a variety of threat-related processes (*Faull et al., 2016*). In this way, rapid threat probability estimates generated by vlPAG Onset neurons may be more akin to brainstem-derived signals for rapid detection of visual threats, which are distributed to a wider brain network (*Liddell et al., 2005*).

Neurons responsive toward the end of cue presentation were more heterogeneous, in terms of their baseline firing rate and their signaling. Ramping neurons prioritized threat probability but also signaled fear output. However, Ramping activity could not drive fear output in full. Differential fear

to safety, uncertainty and danger was observed even in the first second of cue presentation, when these neurons were unresponsive. Unlike Onset neurons, Ramping neurons showed selective firing throughout the delay period, only diminishing once foot shock had been delivered. Ramping neurons may provide a threat probability estimate that increases as threat draws nearer and peaks when threat is imminent. Ramping neurons may help sustain threat estimates in the absence of explicit stimuli, such as in trace conditioning (*McEchron et al., 1998*; *Büchel et al., 1999*), or estimate more precisely *when* the noxious event will occur. Shifts toward PAG-centric activity are apparent in humans, as capture becomes imminent (*Mobbs et al., 2007*) or natural threats draw closer *Mobbs et al., 2010*; with the caveat that neither of these studies could specify the PAG subregion activated.

Ramping neurons may also provide two forms of feedback: the estimated probability of foot shock and a readout of fear output on that trial. Either type of information could be compared to that trial's outcome, particularly for uncertainty trials on which shock can be present or absent. This could be used to adjust estimates of threat probability and fear output on *future* encounters with the cue. This is broadly consistent with a central role for the vlPAG in feedback processes (*McNally et al., 2011*; *Ozawa et al., 2017*; *Yeh et al., 2018*). Even more, this Ramping signal could alter or reduce foot shock processing through descending control of dorsal horn nociceptive inputs via endogenous opioid circuits. This behavioral phenomenon, conditioned analgesia (*MacLennan et al., 1980*; *Fanselow and Bolles, 1979*; *Chance et al., 1978*), may be related to a more general phenomenon of placebo analgesia. VlPAG activation is consistently observed in studies of placebo analgesia (*Tracey, 2010*; *Eippert et al., 2009*; *Wager and Atlas, 2015*; *Petrovic et al., 2002*) and our observation of increasing neural activity toward presentation of a noxious event may provide a suitable neural substrate for this process. Perhaps most remarkable is that although independent, between the Onset and Ramping populations, the vlPAG contains an estimate of threat probability from the time of first encounter up through the noxious event itself. The vlPAG may not signal fear output per se, but is rich with information that would inform a variety of fear processes and behaviors.

If fear output via conditioned suppression is non-linear, and vlPAG activity scales linearly to threat probability, how does the vlPAG fit into the fear circuit? An ultimate explanation for non-linear fear output may be that threat systems evolved to avoid predation, not to precisely match degree of defensive behavior to threat probability. Erring on the side of greater fear to uncertain threats may promote survival. A proximate explanation may be that fear output is the summed product of multiple threat signals. VlPAG output to the RVM may instruct fear output to match threat probability. If this were the only threat signal, then vlPAG activity would in fact reflect fear output. We speculate that additional threat signals govern fear output. For example, neurons in the retrorubral field (RRF) project to the RVM, as well as the CeA (*Zahm and Trimble, 2008*; *Deutch et al., 1988*; *Von Krosigk and Smith, 1991*). Preliminary data from our laboratory suggest that RRF neurons are primarily responsive to danger and uncertainty, but weight uncertainty similarly to danger (danger = uncertainty > safety). RRF neurons may signal threat probability plus uncertainty-induced stress, or may favor cue-shock contiguity over contingency (*Rescorla, 1967*). Summation of vlPAG-derived and RRF-derived threat signals by RVM neurons would produce a non-linear fear output like that observed in our discrimination procedure.

The results pose questions about the specific relationship between the vlPAG and the CeA. VlPAG threat probability signals may be trained up by the CeA, but become CeA-independent with sufficient training (*Ozawa et al., 2017*). Consistent with this interpretation, the CeA is essential to the acquisition of conditioned suppression with limited training, but that more extended training mitigates the effects of CeA lesions (*Lee et al., 2005*; *McDannald, 2010*). This is not to say that we would expect the CeA to become inessential following extensive fear discrimination training. Updating threat probability should occur when environmental stimuli become more or less predictive of noxious events. We anticipate the CeA is essential to updating vlPAG threat probability signaling (*McNally et al., 2011*; *Ozawa et al., 2017*).

It is near universally accepted that the amygdala is a key node of dysfunction in stress (*Rauch et al., 2000*) and anxiety disorders (*Etkin and Wager, 2007*). This may be driven in part by technical considerations: whole-brain fMRI can detect amygdala BOLD signals (*Johnstone et al., 2005*), while detecting subregion-specific PAG BOLD signals requires a more deliberate approach (*Satpute et al., 2013*). Perhaps the primary intellectual driver is that the amygdala is theorized to be

a privileged cite of integration/learning in the fear circuit (*Mahan and Ressler, 2012*; *Admon et al., 2013*). The present findings illustrate that the amygdala is not privileged in this regard, and mark the vlPAG as likely node of dysfunction in psychiatric disorders of stress and anxiety. Appreciation for the vlPAG as a site of integration will hasten mapping of a more complete fear circuit. Deliberate study of vlPAG function (*Arico et al., 2017*; *Assareh et al., 2017*; *Rozeske et al., 2018*) and dysfunction in psychiatric disease (*Yeh et al., 2018*), will be essential to developing effective therapies for disorders characterized by exaggerated threat estimation and aberrant fear.

# Materials and methods

## Key resources table

| Reagent type (species) or resource | Designation | Source or reference | Identifier | Additional information |
|---|---|---|---|---|
| Antibody | Anti-Tryptophan Hydroxylase Raised in Sheep | Sigma | Cat # T8575 | [1:1000] in 0.05M PBS |
| Antibody | Biotinylated Anti-Sheep Raised in Rabbit | Vector Labs | Cat # PK-6106 | [1:200] in 0.05M PBS |
| Chemical compound, drug | Normal Rabbit Serum | Vector Labs | Cat # PK-6106 | 1% in 0.05M PBS |
| Chemical compound, drug | Avidin | Vector Labs | Cat # PK-6106 | [1:200] in 0.05M PBS |
| Chemical compound, drug | Biotin | Vector Labs | Cat # PK-6106 | [1:200] in 0.05M PBS |
| Chemical compound, drug | NovaRED Perioxidase (HRP) Substrate Kit | Vector Labs | Cat # SK-4800 | 18 drops (1), 12 drops (2), 12 drops (3) and 12 drops $H_2O_2$ solution in DI $H_2O$. |
| Chemical compound, drug | Triton | Sigma | Cat # T8787 | |
| Chemical compound, drug | Hydrogen Peroxide | Sigma | Cat # 216763 | |
| Chemical compound, drug | Paraformaldehyde | Sigma | Cat # P6148 | |
| Chemical compound, drug | Sucrose | Fisher Scientific | Cat # S5 | |
| Chemical compound, drug | Sodium Chloride | Fisher Scientific | Cat # S 640 | |
| Chemical compound, drug | Histoprep 100% Reagent Alcohol | Fisher Scientific | Cat # HC800 | |
| Chemical compound, drug | Histoprep 95% Reagent Alcohol | Fisher Scientific | Cat # HC1300 | |
| Chemical compound, drug | Histoclear II | Fisher Scientific | Cat # 5089990150 | |
| Chemical compound, drug | Omnimount | Fisher Scientific | Cat # 5089990146 | |
| Chemical compound, drug | 10% Neutral Buffered Formalin | Fisher Scientific | Cat # 22899402 | |
| Chemical compound, drug | Potassium Phosphate Monobasic | Fisher Scientific | Cat # P285 | |

*Continued on next page*

*Continued*

| Reagent type (species) or resource | Designation | Source or reference | Identifier | Additional information |
|---|---|---|---|---|
| Chemical compound, drug | Potassium Phosphate Dibasic | Fisher Scientific | Cat # P288 | |
| Software, algorithm | MED PC-IV | Med Associates | RRID:SCR_012156 | |
| Software, algorithm | OmniPlex | Plexon | | |
| Software, algorithm | Offline Sorter V6 | Plexon | RRID:SCR_000012 | |
| Software, algorithm | NeuroExplorer | Plexon | RRID:SCR_001818 | |
| Software, algorithm | MATLAB | Mathworks | RRID:SCR_001622 | |
| Software, algorithm | Statistica | StatSoft | RRID:SCR_014213 | |
| Software, algorithm | SPSS | IBM | RRID:SCR_002865 | |
| Software, algorithm | Adobe Illustrator | Adobe | RRID:SCR_010279 | |
| Software, algorithm | Adobe Photoshop | Adobe | RRID:SCR_014199 | |
| Other | Plexon standard commutator | Plexon | Cat # 50122 | |
| Other | Plexon head stage cable – Metal Mesh | Plexon | Cat # 91809–017 | |
| Other | Plexon head stage | Plexon | Cat # 40684–020 | |
| Other | Omnetics connector | Omnetics Corporation | Cat # A79042-001 | |
| Other | Green board - Moveable Array | San Francisco Circuits | Cat # PCB | |
| Other | Stainless Steel ground wire | AM Systems | Cat # 791400 | |
| Other | Formvar Insulated Nichrome Wire | AM Systems | Cat # 761500 | |
| Other | Dustless Precision Pellets | Bio-Serv | Cat # F0021 | |

## Experimental subjects

Ten adult male Long Evans rats (RRID:RGD_2308852) weighing 241–268 g arrived from Charles River Laboratories, Raleigh, NC on postnatal day 55. All rats were implanted with drivable microelectrode bundles. Data are reported from six rats; three rats did not yield units and one rat had incorrect electrode placement. All rats were single-housed throughout the duration of the experiment on a 12 hr light cycle (lights off at 6:00pm) and maintained at 85% of their free-feeding body weight with standard laboratory chow (18% Protein Rodent Diet #2018, Harlan Teklad Global Diets, Madison, WI) except during an 11 day, post-surgery recovery period where animals had *ad libitum* access to standard chow. *Ad libitum* access to water was always available in the home cage. All protocols were approved by the Boston College Animal Care and Use Committee and all experiments were carried

out in accordance with the NIH guidelines regarding the care and use of rats for experimental procedures.

### Electrode assembly

Microelectrodes consisted of a drivable bundle of sixteen 25.4 µm diameter Formvar-Insulated Nichrome wires (761500, A-M Systems, Carlsborg, WA) within a 27-gauge cannula (B000FN3M7K, Amazon Supply) and two 127 µm diameter PFA-coated, annealed strength stainless-steel ground wires (791400, A-M Systems, Carlsborg, WA). All wires were electrically connected to a nano-strip omnetics connector (A79042-001, Omnetics Connector Corp., Minneapolis, MN) on a custom 24-contact, individually routed and gold immersed circuit board (San Francisco Circuits, San Mateo, CA).

### Surgery

Stereotaxic surgery was performed aseptic conditions under isoflurane anesthesia (1–5% in oxygen). Carprofen (i.p., 5 mg/kg) and lactated ringers solution (~2–5 mL) were administered preoperatively. The skull was scoured in a crosshatch pattern with a scalpel blade to increase efficacy of implant adhesion. Five screws were installed in the skull to further stabilize the connection between the skull, electrode assembly and a protective head cap (screw placements: two anterior to bregma, two between bregma and lambda about ~3 mm medial to the lateral ridges of the skull, and one on the midline ~5 mm posterior of lambda). A 1.4 mm diameter craniotomy was performed to remove a circular skull section centered on the implant site and the underlying dura was removed to expose the cortex. Nichrome recording wires were freshly cut with surgical scissors to extend ~2.0 mm beyond the cannula at a ~15° angle. Just before implant, current was delivered to each recording wire in a saline bath, stripping each tip of its formvar insulation. Current was supplied by a 12 V lantern battery and each Omnetics connector contact was stimulated for 2 s using a resistor-equipped lead. Machine grease was placed by the cannula and on the microdrive.

For implantation dorsal to the vlPAG, the electrode assembly was slowly advanced at a 20° angle to the following coordinates from cortex (anterior-posterior: −8.00 mm, medial-lateral: −2.45 mm and dorsal-ventral: −5.52 mm). Once in place, stripped ends of both ground wires were wrapped around a sixth screw inserted previously to ground the electrode (anterior-posterior: −8.00 mm, medial-lateral: +2.45 mm). The microdrive base and a protective head cap surrounding the electrode assembly were cemented in place at the end of the procedure using orthodontic resin (C 22-05-98, Pearson Dental Supply, Sylmar, CA).

### Behavior apparatus

The apparatus for Pavlovian fear conditioning consisted of two individual chambers with aluminum front and back walls retrofitted with clear plastic covers, clear acrylic sides and top, and a grid floor. Each grid floor bar was electrically connected to an aversive shock generator (Med Associates, St. Albans, VT) through a grounding device. This permitted the floor to be grounded at all times except during shock delivery. An external food cup and a central nose poke opening, equipped with infrared photocells were present on one wall. Auditory stimuli were presented through two speakers mounted on the ceiling.

### Nose poke acquisition

Prior to discrimination sessions, rats were food-deprived to 85% of their free-feeding body weight and were fed specifically to maintain this weight through the behavioral procedure. Starting on P59, rats were shaped to nose poke for pellet delivery in the experimental chamber using a fixed ratio schedule in which one nose poke yielded one pellet. Shaping sessions lasted 30 min or until approximately 50 nose pokes were completed. Over the next 3 days, rats were placed on 5 days of variable interval (VI) schedules in which nose pokes were reinforced on average every 30 s (day 1), or 60 s (days 2 through 5). For the remainder of behavioral testing, nose pokes were reinforced on a VI-60 schedule independent of all Pavlovian contingencies.

## Pre-exposure

In two separate sessions, each rat was pre-exposed to the three cues to be used in Pavlovian discrimination. Auditory cues consisted of repeating motifs of broadband click, phaser or trumpet. These 42 min sessions consisted of four presentations of each cue (12 total presentations) with a mean inter-trial interval (ITI) of 3.5 min. The order of trial type presentation was randomly determined by the behavioral program and differed for each rat during each session.

## Fear discrimination

Prior to recording, each rat received eight, 93 min sessions of fear discrimination. Each session consisted of 32 trials, with a mean ITI of 3.5 min. Auditory cues were 10 s in duration and consisted of repeating motifs of a broadband click, phaser, or trumpet. Each cue was associated with a unique probability of foot shock (0.5 mA, 0.5 s): danger, p=1.00; uncertainty, p=0.375; and safety, p=0.00. Auditory identity was counterbalanced across rats. Foot shock was administered 2 s following the termination of the auditory cue on danger and uncertainty shock trials. This was done in order to observe possible neural activity during the delay period not driven by an explicit cue. A single session consisted of six danger trials, ten uncertainty no-shock trials, six uncertainty shock trials, and ten safety trials. The order of trial type presentation was randomly determined by the behavioral program, and differed for each rat, each session. After the eighth session, rats were removed from discrimination, given full food and received stereotaxic surgery. Following recovery, discrimination (identical to that described above) resumed with single-unit recording. Animals received discrimination every other day with recording. After each discrimination session with recording, electrodes were advanced either 0.042 mm or 0.084 mm to record from new units during the following session.

## Histology

Rats were deeply anesthetized using isoflurane and final electrode coordinates were marked by passing current from a 6 V battery through 4 of the 16 nichrome electrode wires. Rats were perfused with 0.9% biological saline and 4% paraformaldehyde in a 0.2 M Potassium Phosphate Buffered Solution. Brains were extracted and post-fixed in a 10% neutral-buffered formalin solution for 24 hr, stored in 10% sucrose/formalin and sectioned via microtome. All brains processed for light microscopy using anti-tryptophan hydroxylase immunohistochemistry (T8575, Sigma-Aldrich, St. Louis, MO) and a NovaRed chromagen reaction (SK-4800, Vector Laboratories, Burlingame, CA) Sections were mounted, imaged using a light microscope and electrode placement was confirmed (*Paxinos and Watson, 2007*).

## Single-unit data acquisition

Sixteen individual recording wires were bundled and soldered to individual channels of an Omnetics connector. The bundle was integrated into a microdrive permitting advancement in ~0.042 mm increments. The microdrive was cemented on top of the skull and the Omnetics connector was affixed to the head cap. During recording sessions, a 1x amplifying head stage connected the Omnetics connector to the commutator via a shielded recording cable (head stage: 40684–020 and Cable: 91809–017, Plexon Inc, Dallas TX). Analog neural activity was digitized and high-pass filtered via amplifier to remove low-frequency artifacts and sent to the Ominplex D acquisition system (Plexon Inc, Dallas TX). Behavioral events (cues, shocks, nose pokes) were controlled and recorded by a computer running Med Associates software. Timestamped events from Med Associates were sent to Ominplex D acquisition system via a dedicated interface module (DIG-716B). The result was a single file (.pl2) containing all time stamps for recording and behavior. Single-units were sorted offline with a template-based spike-sorting algorithm (Offline Sorter V3, Plexon Inc, Dallas TX). Timestamped spikes and events (cues, shocks, nose pokes) were extracted and analyzed with statistical routines in MATLAB (Natick, MA). Neural activity was recorded throughout the 500 ms shock delivery period. However, we cannot be certain that shock artifact did not disrupt spike collection, so we do not present activity from this period.

## Statistical analysis

### Calculating suppression ratios

Fear was measured by suppression of rewarded nose poking, calculated as a ratio: (baseline poke rate – cue poke rate) / (baseline poke rate +cue poke rate) (*Rescorla, 1968*; *Pickens et al., 2009*; *Anglada-Figueroa and Quirk, 2005*; *Arico and McNally, 2014*; *Lee et al., 2005*; *McDannald and Galarce, 2011*). A ratio of '1' indicated high fear, '0' low fear, and gradations between intermediate levels of fear. Use of the suppression ratio permitted the objective measure of relative fear in 1 s intervals across the cue, as well as total fear over the entire 10 s cue presentation (*Wright et al., 2015*).

### K-means clustering

The following characteristics were determined for each neuron: baseline firing rate, half the duration of the mean waveform and amplitude ratio of the mean waveform. Duration was determined by measuring the time (ms) from peak depolarization to the trough of after-hyperpolarization and dividing by two. Amplitude ratio was calculated using (n – p) / (n + p), in which p=initial hyperpolarization (in mV) and n = maximal depolarization (in mV). This approach has been used to successfully separate neuron types in the ventral tegmental area (*Roesch et al., 2007*). K-means clustering used these three firing characteristics to partition the 245 recorded neurons into two clusters (k = 2). Two clusters were chosen because previous studies have found that two neuron types, glutamatergic vGluT2 neurons and GABAergic Gad1 neurons, comprise the majority of vlPAG neurons, and these neurons can be differentiated by baseline firing rate (*Tovote et al., 2016*). ANOVA for cluster results found that only baseline firing rate contributed to cluster membership ($F_{1,243}$ = 829, p<0.001). Neither amplitude ratio nor duration reached significance (Fs <0.2, ps >0.6). All neurons were clustered, with the majority falling in the low firing rate cluster (n = 199) and the remaining in the high firing rate cluster (n = 46).

### Identifying cue-responsive vlPAG neurons

Independent of cluster analysis, all 245 neurons were screened for short-latency, excitatory firing to auditory cue onset (*Tovote et al., 2016*; *Watson et al., 2016*; *Ozawa et al., 2017*). This was achieved using a paired, two-tailed t-test comparing raw firing rate (spikes/s) during a 2 s baseline period just prior to cue onset and during the first, 1 s cue interval. A t-test was performed for each of the three cues (danger, uncertainty and safety), corrected for multiple comparisons (p<0.017). The remaining neurons were screened for longer-latency, excitatory firing to the later portion of auditory cues (*Ozawa et al., 2017*), using an identical t-test, only now comparing firing rate during a 2 s baseline period just prior to cue onset and the last, 1 s cue interval.

### Z-score normalization

For each neuron, and for each trial type, firing rate (spikes/s) was calculated in 100 ms bins from 10 s prior to cue onset to 12 s following cue offset, for a total of 320 bins. Mean firing rate over the 320 bins was calculated by averaging all trials for each trial type. Mean differential firing was calculated for each of the 320 bins by subtracting mean baseline firing rate (2 s prior to cue onset), specific to that trial type, from each bin. Mean differential firing was Z-score normalized across all trial types within a single neuron, such that mean firing = 0, and standard deviation in firing = 1. Z-score normalization was applied to firing across the entirety of the recording epoch, as opposed to only the baseline period, in case neurons showed little/no baseline activity. As a result, periods of phasic, excitatory firing contributed to normalized mean firing rate (0). For this reason, Z-score normalized baseline activity is below zero in *Figure 2A & C*. Z-score normalized firing during cue (*Figure 2A,C*) and post-cue periods (*Figure 6A,B*), was analyzed with ANOVA using bin and trial-type as factors. F and p values are reported, as well as partial eta squared and observed power.

For post hoc cue firing analyses (*Figure 2B,D*), cue correlation analyses (*Figure 3* all), and cue regression analyses, it was necessary to calculate normalized firing in 1 s intervals. To do this, differential firing in the interval of interest (for example, first cue 1 s interval) was calculated for each individual of the 32 trials in a single session. Differential firing in this interval was then Z-score transformed. This process was repeated for each interval of interest. This done in order to maximize the distribution of firing within a single interval. Importantly, statistical outcomes were identical if a

single Z-score transformation was applied to all intervals at once. An identical approach was used to Z-score normalize firing in 500 ms intervals for post-cue firing (*Figure 6*).

### Determining observed and expected cue firing patterns

The analysis for Onset neurons (n = 29) utilized mean normalized firing to each cue (danger, uncertainty and safety) in the first 1 s interval; analysis for Ramping neurons (n = 14) utilized firing in the last 1 s interval. Relative firing to the three cues was used to categorize each Onset and Ramping neuron: (d > u > s), (d > s > u), (s > u > d) or (u > d > s). Counting the number observed in each category determined the actual number for each population. In order to determine the number in each category expected by chance, each neurons firing pattern was shuffled and the number of neurons in each category counted. Shuffling and counting neuron category was repeated 1000 times for each population. Box and whisker plots for each category/population were constructed showing the median, 25th percentile, 75th percentile and most extreme non-outliers. The actual number and expected statistics are reported plotted together in *Figure 4D/H*.

### Population and single-unit firing analyses

Population firing was analyzed using analysis of variance (ANOVA) with trial type and bin (100 ms) as factors. ANOVA for cue firing contained three trial types (danger, uncertainty and safety). Uncertainty trial types were collapsed because they did not differ for either suppression ratio or firing analysis. This was expected, during cue presentation rats did not know the current uncertainty trial type. ANOVA for post-cue firing contained four trial types (danger, uncertainty shock, uncertainty omission and safety). Uncertainty trial types were split because shock was delivered during the period. F statistic, p value, observed power and partial eta squared are reported for effects and interactions. Interval firing was compared within a population using a two-tailed, dependent samples t-test. Identical firing analyses were performed for the post-cue period, only now four trial types were used (danger, uncertainty shock, uncertainty omission and safety).

The sliding window analysis for the Ramping population (*Figure 2E*) employed a two-tailed, dependent samples t-test. Ramping population danger firing was compared to safety firing in 1 s intervals, with the window advancing in 100 ms steps across cue presentation. The p value for each comparison was recorded, and the first window with p<0.05 was termed the interval of departure (when danger and safety firing significant differed). The same analysis was performed for uncertainty and safety. Change in firing from interval of departure to the final 1 s window was calculated by subtracting the departure firing rate from the last 1 s window firing rate and dividing by time between these two periods.

Biases in single-unit firing to the three cues (*Figure 3*) were determined using the sign test comparing z firing to danger vs. uncertainty, and uncertainty vs. safety. The relationship between firing was determined using the Pearson correlation coefficient.

### Single-unit, linear regression

Single-unit, linear regression was used to determine the degree to which fear output and threat probability explained trial-by-trial variation in firing of single neurons in a specific time interval. The cue analysis used 1 s intervals, while the post-cue analyses used 500 ms intervals. Shorter intervals were used in the post-cue analysis to accommodate the 500 ms foot shock duration. For each regression, all 32 trials from a single session were ordered by type. Z-firing was specified for the interval of interest. The interval fear regressor was the suppression ratio for that specific interval/trial, while the total fear regressor was the suppression ratio for the entire cue, for that specific trial. The probability regressor was the foot shock probability associated with the specific cue. Regression (using the regress function in MATLAB) required a separate, constant input. To better visualize the organization of the regression input, the complete regression input for first interval firing of an Onset neuron is shown below. Identical regression analysis was performed for the post-cue period, only now 500 ms intervals were examined and the total fear regressor was removed. This interval was used to accommodate the 500 ms shock period and the total fear regressor was removed because it only sampled behavior from the 10 s cue period.

| trial # | trial type | Z firing | constant | interval fear | total fear | probability |
|---|---|---|---|---|---|---|
| 1 | danger | 2.32 | 1 | 1.00 | 1.00 | 1 |
| 2 | danger | 0.51 | 1 | 1.00 | 1.00 | 1 |
| 3 | danger | 0.81 | 1 | 1.00 | 1.00 | 1 |
| 4 | danger | 1.56 | 1 | 1.00 | 1.00 | 1 |
| 5 | danger | 1.41 | 1 | -1.00 | 0.73 | 1 |
| 6 | danger | 2.92 | 1 | 1.00 | 1.00 | 1 |
| 7 | unc-shock | 0.81 | 1 | 1.00 | 1.00 | 0.375 |
| 8 | unc-shock | -0.40 | 1 | -0.33 | 0.47 | 0.375 |
| 9 | unc-shock | 0.21 | 1 | 1.00 | 0.64 | 0.375 |
| 10 | unc-shock | -0.70 | 1 | 1.00 | 0.67 | 0.375 |
| 11 | unc-shock | -0.85 | 1 | 1.00 | 1.00 | 0.375 |
| 12 | unc-shock | -0.70 | 1 | -1.00 | 0.64 | 0.375 |
| 13 | unc-omission | -0.70 | 1 | 1.00 | 1.00 | 0.375 |
| 14 | unc-omission | -0.70 | 1 | -1.00 | 0.40 | 0.375 |
| 15 | unc-omission | -0.70 | 1 | 0.00 | 0.23 | 0.375 |
| 16 | unc-omission | 0.66 | 1 | 1.00 | 1.00 | 0.375 |
| 17 | unc-omission | 0.21 | 1 | 1.00 | 1.00 | 0.375 |
| 18 | unc-omission | -0.70 | 1 | 1.00 | 0.71 | 0.375 |
| 19 | unc-omission | -1.00 | 1 | 1.00 | 0.43 | 0.375 |
| 20 | unc-omission | 1.11 | 1 | 1.00 | 1.00 | 0.375 |
| 21 | unc-omission | -0.09 | 1 | -0.60 | 0.56 | 0.375 |
| 22 | unc-omission | -0.85 | 1 | 1.00 | 1.00 | 0.375 |
| 23 | safety | -0.70 | 1 | -0.33 | 0.06 | 0 |
| 24 | safety | -0.70 | 1 | 0.00 | 0.27 | 0 |
| 25 | safety | -0.70 | 1 | 1.00 | -0.04 | 0 |
| 26 | safety | -0.40 | 1 | 1.00 | 0.14 | 0 |
| 27 | safety | -0.70 | 1 | 1.00 | 0.54 | 0 |
| 28 | safety | -0.70 | 1 | -0.50 | 0.03 | 0 |
| 29 | safety | 0.21 | 1 | 1.00 | 0.47 | 0 |
| 30 | safety | -0.85 | 1 | 1.00 | 1.00 | 0 |
| 31 | safety | 0.06 | 1 | 1.00 | 0.20 | 0 |
| 32 | safety | -0.70 | 1 | 1.00 | -0.14 | 0 |
| | | | beta coefficient: | 0.24 | -0.14 | 2.19 |

The regression output of greatest interest was the beta coefficient for each regressor (interval fear, total fear and probability), quantifying the strength (greater distance from zero = stronger) and direction (>0 = positive) of the predictive relationship between each regressor and single-unit firing. ANOVA, two-tailed dependent samples t-test, sign test and The Pearson correlation coefficient was used to analyze beta coefficients, exactly as described for normalized firing. An identical approach was used for the pairs analysis (*Figure 3*) only now firing of the Ramping neuron was used as a regressor to predict firing of the paired Onset neuron. Beta coefficient (β) and significance of the predictive relationship (*p*) are reported.

## Threat probability tuning curve

Single-unit, linear regression was performed using the interval fear, total fear and probability regressor as above. Only now, nine separate regression analyses were performed in which the uncertainty component of the probability regressor was systematically varied from 0 to 1 in 0.125 increments

(0.000, 0.125, 0.250, 0.375, 0.500, 0.625, 0.750, 0.875 and 1.000). The result of primary interest was the mean beta coefficient for the probability regressor from each variant of regression, as plotted in *Figure 5E/J*.

### Correlation matrix

A correlation matrix was created for the post-cue period for the Onset population (n = 29) as well as the Ramping population (n = 14). The periods of interest were the four, 500 ms intervals prior to shock presentation and the five, 500 ms intervals following shock. Starting with interval 1, The Pearson correlation coefficient ($R^2$) and associated p value was calculated by comparing the probability beta coefficient for each neuron to those for intervals 2–4 and 6–10. Most critical were the comparison with intervals 6–10, that permitted comparison of beta coefficients between pre-shock and post-shock periods. The same relationships were determined for each of the remaining intervals (2–4, and 6–10). The p value of $R^2$ for each interval comparison is plotted. The proportion of significant intervals forming the pre-shock and post-shock comparison (n = 20) were compared for the Onset and Ramping populations using the chi-square test.

### Data and software availability

Full electrophysiology data set will be uploaded to http://crcns.org/ upon acceptance for publication.

### Additional resources

Med Associates programs used for behavior and MATLAB programs used for behavioral analyses are made freely available at our lab website: http://mcdannaldlab.org/resources.

## Acknowledgements

We thank Dr. Hiram Brownell for statistical advice on linear regression, Dr. Thomas C Jhou for designing and assembling the shock floor grounding device, Dr. Donald B Katz for guidance on electrode construction, Dr. Thomas Stalnaker for guidance on waveform analysis, Blake Zimmerman for guidance on MATLAB scripts, Jasmin Strickland for manuscript feedback and Daniel Pimpinelli for assistance with animal care. This work was supported by NIH DA034010 to MAM.

## Additional information

### Funding

| Funder | Grant reference number | Author |
|---|---|---|
| National Institutes of Health | MH117791 | Michael A McDannald |
| National Institutes of Health | DA034010 | Michael A McDannald |

The funders had no role in study design, data collection and interpretation, or the decision to submit the work for publication.

### Author contributions

Kristina M Wright, Conceptualization, Data curation, Formal analysis, Investigation, Visualization, Methodology, Writing—original draft, Writing—review and editing; Michael A McDannald, Conceptualization, Resources, Data curation, Software, Formal analysis, Supervision, Funding acquisition, Validation, Investigation, Visualization, Methodology, Writing—original draft, Project administration, Writing—review and editing

### Author ORCIDs

Kristina M Wright  http://orcid.org/0000-0003-1446-3009
Michael A McDannald  http://orcid.org/0000-0001-8525-1260

## Ethics

Animal experimentation: This study was performed in strict accordance with the recommendations in the Guide for the Care and Use of Laboratory Animals of the National Institutes of Health. All of the animals were handled according to approved institutional animal care and use committee (IACUC) protocols (#2018-002) of Boston College. All surgery was performed under isofluorane anesthesia, and every effort was made to minimize suffering.

## Decision letter and Author response

Decision letter https://doi.org/10.7554/eLife.45013.023
Author response https://doi.org/10.7554/eLife.45013.024

## Additional files

### Supplementary files

• Transparent reporting form
DOI: https://doi.org/10.7554/eLife.45013.019

### Data availability

Single-unit data are publicly available on CRCNS (http://crcns.org/data-sets/brainstem/pag-1), under the doi: 10.6080/K0R49P0V. Users must first create a free account (https://crcns.org/register) before they can download the datasets from the site.

The following dataset was generated:

| Author(s) | Year | Dataset title | Dataset URL | Database and Identifier |
| --- | --- | --- | --- | --- |
| Wright KM, McDannald MA | 2019 | Data from: Ventrolateral periaqueductal gray neurons prioritize threat probability over fear output | http://dx.doi.org/10.6080/K0R49P0V | CRCNS, 10.6080/K0R49P0V |

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
