## [Decision Letter]

Thank you for submitting your article "Ventrolateral periaqueductal gray neurons prioritize threat probability over fear output" for consideration by *eLife*. Your article has been reviewed by three peer reviewers, including Geoffrey Schoenbaum as the Reviewing Editor, and the evaluation has been overseen by Laura Colgin as the Senior Editor. The following individuals involved in review of your submission have agreed to reveal their identity: Satoshi Ikemoto (Reviewer #2) and Laura Bradfield (Reviewer #3).

The reviewers have discussed the reviews with one another and the Reviewing Editor has drafted this decision to help you prepare a revised submission.

Summary:

In this study the authors recorded from PAG neurons during a fear discrimination task. Three cues predicted certain, uncertain, and no shock. They measured fear during the 10s cues as suppression behavior, demonstrating discriminative fear responding. However the degree of the difference between the certain and uncertain shock was larger than the difference in suppression, especially early, which was quite similar for these two cues. Subsequent analyses of unit firing revealed populations of onset and ramping neurons that seemed to reflect the fear response as shown previously. However a more careful analysis regressing firing on probability of shock versus behavior showed that onset neurons were primarily reflecting probability and ramping neurons reflected a combination of the two. These data nicely show that contrary to current proposals in simpler tasks, the firing of PAG neurons is not simply reflective of behavioral or fear output and instead reflects a more careful assessment of the actual probability of shock or danger.

Essential revisions:

Overall the reviewers were very positive. The report provides novel information on how PAG responses correlate with shock probability versus behavioral output, information somewhat contradictory to current ideas that PAG drives the response. However all three reviewers shared the general concern that the conclusion did not consider some possible alternatives or problems in the approach. Reviewer #1 suggests that it assumes a linear relationship between shock probability and behavior; reviewer #2 worries that there is not a sufficient exploration of the probability space for a regression analysis, and reviewer #3 raises the possibility that other factors, like stress, might add to the fear from shock probability. These concerns might be addressed with additional analyses (population comparisons or analyses of specific units showing that many discriminate different probability cues when behavior does not). However, to some extent they mostly require some mention in the Discussion. Being clear that suppression is what is measured (in Abstract and elsewhere) as fear may also be helpful. In addition to this, reviewer #3 also made the point that the recording study is correlational. Some discussion regarding whether these signals are generated in PAG or received from upstream areas like amygdala, and whether they are prominent enough to guarantee output downstream would be a useful addition.

Minor points:

Please look through the paper and try to smooth out any complex writing like that identified by reviewer 2. Please see the individual reviews, which are provided in their entirety below, for other suggestions.

*Reviewer #1:*

In this study the authors recorded from PAG neurons during a fear discrimination task. Three cues predicted certain, uncertain, and no shock. They measured fear during the 10s cues as suppression behavior, demonstrating discriminative fear responding. However the degree of the difference between the certain and uncertain shock was larger than the difference in suppression, especially early, which was quite similar for these two cues. Subsequent analyses of unit firing revealed populations of onset and ramping neurons that seemed to reflect the fear response as shown previously. However a more careful analysis regressing firing on probability of shock versus behavior showed that onset neurons were primarily reflecting probability and ramping neurons reflected a combination of the two. These data nicely show that contrary to current proposals in simpler tasks, the firing of PAG neurons is not simply reflective of behavioral or fear output and instead reflects a more careful assessment of the actual probability of shock or danger. Overall I thought the study was creative and the analysis was careful and the results clear. I have only a few suggestions.

1) The initial screen seemed to identify neurons that had heightened activity to any of the cues. Were all the neurons selective for the proper order or was there variability in preferred cues?

I think the supplementary figure showing the variance captured by different regressors was convincing. Some display of this might be in the main text.

One alternative or clarification that I think might be helpful also is for the authors to more clearly explain how their task allows them to dissociate the two ideas about what PAG neurons are encoding. Assuming I am correct this is both simple as well as an elegant solution to the problem. It should be made clear I think to help the reader understand how it is done. Relatedly a small caveat may need to be put in the Discussion to the effect that their conclusion (I think) assumes a linear relationship between firing and behavior. Since they have another variable – threat – that provides a much better correlation I accept this as a reasonable assumption here. But I think if I am correct that it should be mentioned.

*Reviewer #2:*

The authors investigated neural activity of the vlPAG with respect to behavioral suppression induced by shock-associated cues. They focused on two types of neurons: ones that responded to the onset of cue associated with shock and those that displayed ramping responses to shock. They found that activities of both types of neurons were better correlated with the probability of shock delivery rather than cue-induced suppression.

My concern is that a key finding of the study depends on a linear regression analysis containing a measurement variable with only three values: threat probabilities of 0, 0.375, and 1.0. I would have felt more confident if the experiment had more probability values.

*Minor Comments:*

Writing is hard. It sometimes requires effort to understand. Here are examples:

“ramping activity prioritized threat probability over fear output.”

“vlPAG signals for fear output may reside in other functional populations, such has neurons showing inhibition of firing to danger cues [Tovote, Fadok and Luthi, 2015].”

“Yet, this would still require that adjacent populations for threat probability and fear output exist within the vlPAG.”

Abstract

The key dependent measure, conditioned suppression, should be mentioned in the Abstract section. Remember that this is a primary scientific report. Although theoretical constructs such as "fear" can be mentioned, the concrete measure of fear should be identified in Abstract.

Results

Throughout the Results section, t-tests were conducted as post-hoc comparisons after ANOVAs. Are the familywise error rates controlled?

Discussion

An observation with statistical data is described for the first time in the Discussion section. Such information should be reported in the result section. In addition, the authors discussion is not coherent. The authors first state that their electrophysiological observation was performed in the caudal vlPAG. Then, they state that the neural activity of the caudal vlPAG is not different from those of the rostral subregion. Explain how they can tell subregional difference when they focused on the caudal subregion.

Perhaps, it is important to discuss the present findings in light of the findings that selective lesions of vlPAG is sufficient to disrupt freezing. It does not seem to be difficult to explain how the loss of neurons coding threat probability can disrupt behavioral expression of fear.

The authors state "We did not observe a substantial population of vlPAG neurons with these temporal firing characteristics (Figure 1—figure supplement 3), making the vlPAG a poor candidate for signaling most forms of sustained fear output". This statement is based on the assumption that much larger neural populations are needed to control behavior. However, research on neural ensemble suggests that the loss of only a handful neurons is sufficient to affect behavior. For example, Koya et al., 2009, found that only 2 – 3% of neurons are responsible for mediating context-specific sensitization induced by cocaine administration (Nat Neurosci, 12(8), 1069-1073. doi: http://www.nature.com/neuro/journal/v12/n8/suppinfo/nn.2364_S1.html).

*Reviewer #3:*

Wright and McDannald present a comprehensive and elegant set of experiments showing that, contrary to the currently prevailing view, single-unit signalling in the ventrolateral periaqueductal gray (vlPAG) appears to reflect threat probability independently of fear output. They used three distinct auditory cues that predicted footshock with certainty (p = 1), uncertainty (p =.375) or not at all (i.e. safety, p = 0). They found that two distinct populations of vlPAG neurons: onset and ramping neurons, primarily reflected threat probability rather than fear output (as measured by conditioned suppression of an appetitively-trained nose-poke behaviour). Overall I think this article is superb, extremely well-written and easy to follow, a very interesting topic, and a thoughtful discussion of results.

*Minor Comments:*

There are a couple of very minor points I would like to suggest for improvement. None of the concerns are substantive.

1) The most pressing issue I would like to see addressed is the fact that although electrophysiological recording of single-units is extremely informative, it does not actually provide us with causal data, only correlational. Indeed, the authors acknowledge and deal with this in the discussion, there addressing most of the notes I had made to myself whilst reading the manuscript, which was very satisfying. However, there's one more possibility I would like to see addressed, and that is that the firing patterns they are observing in vlPAG neurons represent information being received from the amygdala, rather than a signal that is being transferred to downstream targets. In other words, does the amygdala code for threat probability, send this information to vlPAG where it then translates it into the necessity for a particular fear response. Unfortunately this is impossible to test with ephys, but I would like to see it mentioned as a possibility, and possibly argued against if the authors see it as unlikely.

2) Further, it is worth noting that uncertainty in and of itself is a stressor, which could explain the fact that the fear output towards the uncertainty cue is higher than might be expected. It could be that the amygdala is encoding the threat probability and that information is being conveyed to the vlPAG, but that the additional stress from uncertainty is being encoded elsewhere in the brain and added to the fear output response at some unknown mediating output structure.

3) In the Results it would be really helpful to see how many rats/neurons etc. were included in the study as I had to search through to find this information.

4) In Figure 1—figure supplement 3, the authors suggest (at least as I understand it) that there is no relation between normalised firing rate and suppression ratio in the overall population of neurons, but from looking at the graph it appears that there might be at least a weak negative correlation? That is, when normalised firing rate was lowest, in the danger condition, suppression ratio appears to be at its highest (indicating high fear), and vice versa for the safety conditions. Is that correct or am I missing something?

5) In the last paragraph of the subsection “Ramping neurons signal threat probability through delay until shock receipt”, Figures 1E and 1F are referred to, I believe this should say Figures 6E and 6F.

---

## [Author Response]

Reviewer #1:

[…] 1) The initial screen seemed to identify neurons that had heightened activity to any of the cues. Were all the neurons selective for the proper order or was there variability in preferred cues?

This is an excellent idea. To address this, we have performed a new analysis to address this question and the results are reported in Figure 4D/H (subsection “Population biases are evident in vlPAG single-units”). Separately for Onset and Ramping populations, we counted the number of neurons showing the ordered firing pattern (danger > uncertainty > safety). We also counted the number of neurons showing the next 3 most common firing patterns. We shuffled each neuron’s normalized firing rate to danger, uncertainty and safety and then counted the number of neurons for each pattern type. This process was repeated 1000 times to generate a distribution from the shuffled firing data. This would tell how often we would expect each pattern to occur by chance. The actual values were then compared to the distributions. The number of Onset and Ramping neurons showing the ordered firing pattern (danger > uncertainty > safety) fell entirely outside of the shuffled distribution, while neurons with all other firing patterns fell within the distribution. The results reveal that the ordered firing pattern was greater than would have been expected by chance.

I think the supplementary figure showing the variance captured by different regressors was convincing. Some display of this might be in the main text.

In order to address a concern of reviewer 2, we performed a new analysis in which we asked if the relative firing pattern to danger, uncertainty and safety by Onset (subsection “Onset neurons signal threat probability”) and Ramping neurons (subsection “Ramping neurons prioritize threat probability over fear output”) was better captured by a probability other than 0.375. The result of this analysis was a ‘tuning curve’ showing that Onset neurons were best captured by the actual shock probability while Ramping neurons were slightly shifted right (now reported in Figure 5). As a part of this analysis, one can visualize the result of the ‘binary’ regression result in which safety and uncertainty were both assigned probabilities of 0.000. We have made sure to point this out in the Results section and have kept the full description of the binary regression in the supplement.

One alternative or clarification that I think might be helpful also is for the authors to more clearly explain how their task allows them to dissociate the two ideas about what PAG neurons are encoding. Assuming I am correct this is both simple as well as an elegant solution to the problem. It should be made clear I think to help the reader understand how it is done. Relatedly a small caveat may need to be put in the Discussion to the effect that their conclusion (I think) assumes a linear relationship between firing and behavior. Since they have another variable – threat – that provides a much better correlation I accept this as a reasonable assumption here. But I think if I am correct that it should be mentioned.

We agree that the non-linear behavior output we observed must be reconciled with the linearly scaled activity we observe in the vlPAG. We have now devoted a significant portion of the Discussion to this problem (tenth-twelfth paragraphs). Briefly, we think that a vlPAG-derived threat signal is but one of multiple signals that ultimately control fear output. We have evidence that a threat signal originating from the retrorubral field weights uncertainty much more like danger. Summing these two threat signals would better approximate the non-linear behavior output we see in our discrimination procedure.

We also agree that understanding this non-linearity is key to understanding the results. For this reason we now also describe the non-linearity of fear output in the Introduction to better set up the reader for the analyses we perform in the manuscript.

Reviewer #2:

[…] My concern is that a key finding of the study depends on a linear regression analysis containing a measurement variable with only three values: threat probabilities of 0, 0.375, and 1.0. I would have felt more confident if the experiment had more probability values.

This is a valid concern. To address this, we have performed a new regression analysis in which we incremented the probability assigned to uncertainty from 0.000 to 1.000 in 0.125. We report the β coefficient for threat probability for each increment in Figure 5E/J (subsection “Ramping neurons prioritize threat probability over fear output”). The result is a ‘tuning curve’ showing the uncertainty probability that best captures the relative firing pattern of each population. For Onset neurons, this analysis revealed the actual probability of 0.375 to be the ‘peak’ of the tuning curve. This provides further evidence that Onset neurons are tuned to shock probability. For Ramping neurons, the peak was shifted right, consistent with more mixed signaling of threat probability and fear output. We are grateful for this suggested analysis as we feel it confirms and extends the regression findings.

Minor Comments:Writing is hard. It sometimes requires effort to understand. Here are examples:“ramping activity prioritized threat probability over fear output.”“vlPAG signals for fear output may reside in other functional populations, such has neurons showing inhibition of firing to danger cues [Tovote, Fadok and Luthi, 2015].”“Yet, this would still require that adjacent populations for threat probability and fear output exist within the vlPAG.”

We agree that those sections, and others, were difficult to read. We have revised them and have carefully read through the remainder of the manuscript. In addition, we have solicited the help of our new postdoc, Jasmin Strickland – someone with extensive knowledge of behavior but not involved in writing this manuscript – to read all sections for clarity. We have revised sections she found difficult to understand. She is acknowledged for her contribution.

AbstractThe key dependent measure, conditioned suppression, should be mentioned in the Abstract section. Remember that this is a primary scientific report. Although theoretical constructs such as "fear" can be mentioned, the concrete measure of fear should be identified in Abstract.

We completely agree. We now describe conditioned suppression as our measure of fear in the Abstract. We also note the experiment was performed with male, Long Evans rats.

ResultsThroughout the Results section, t-tests were conducted as post-hoc comparisons after ANOVAs. Are the familywise error rates controlled?

For the initial submission, familywise error rates were controlled for the majority of analyses. In revising the manuscript, we found that we had not performed these corrections for the paired t-tests in the interval suppression ratio figure (Figure 1D) as well as the single-sample t-tests in the interval regression figures (Figure 5D/I and Figure 6C/D). We have now corrected for multiple tests in all of these figures. The statistical patterns did not change.

DiscussionAn observation with statistical data is described for the first time in the Discussion section. Such information should be reported in the result section. In addition, the authors discussion is not coherent. The authors first state that their electrophysiological observation was performed in the caudal vlPAG. Then, they state that the neural activity of the caudal vlPAG is not different from those of the rostral subregion. Explain how they can tell subregional difference when they focused on the caudal subregion.

This is a fair point. We have moved all stats reported in the Discussion to the Results. We apologize for the poor wording re: rostral and caudal. The vlPAG has an extensive anterior/posterior extent. In comparison to the full extent, all of our recordings came from the caudal portion. Our recordings centered on bregma levels -7.62 and -7.98. We meant to state that there was no difference in the signals obtained from these two recording levels. We also note that the vlPAG extends nearly to -8.70, and that because we did not record from this most extreme caudal extent, it is possible that neurons signaling fear output reside there. We have clarified this point in the Discussion.

Perhaps, it is important to discuss the present findings in light of the findings that selective lesions of vlPAG is sufficient to disrupt freezing. It does not seem to be difficult to explain how the loss of neurons coding threat probability can disrupt behavioral expression of fear.

We definitely agree and have added this to the manuscript (Discussion).

The authors state "We did not observe a substantial population of vlPAG neurons with these temporal firing characteristics (Figure 1—figure supplement 3), making the vlPAG a poor candidate for signaling most forms of sustained fear output". This statement is based on the assumption that much larger neural populations are needed to control behavior. However, research on neural ensemble suggests that the loss of only a handful neurons is sufficient to affect behavior. For example, Koya et al., 2009, found that only 2 – 3% of neurons are responsible for mediating context-specific sensitization induced by cocaine administration (Nat Neurosci, 12(8), 1069-1073. doi: http://www.nature.com/neuro/journal/v12/n8/suppinfo/nn.2364_S1.html).

We agree and have removed all language suggesting the small number of high firing rate cue-responsive neurons diminishes their possible role in behavior.

Minor Comments:There are a couple of very minor points I would like to suggest for improvement. None of the concerns are substantive.1) The most pressing issue I would like to see addressed is the fact that although electrophysiological recording of single-units is extremely informative, it does not actually provide us with causal data, only correlational. Indeed, the authors acknowledge and deal with this in the discussion, there addressing most of the notes I had made to myself whilst reading the manuscript, which was very satisfying. However, there's one more possibility I would like to see addressed, and that is that the firing patterns they are observing in vlPAG neurons represent information being received from the amygdala, rather than a signal that is being transferred to downstream targets. In other words, does the amygdala code for threat probability, send this information to vlPAG where it then translates it into the necessity for a particular fear response. Unfortunately this is impossible to test with ephys, but I would like to see it mentioned as a possibility, and possibly argued against if the authors see it as unlikely.

All of these observations are valid and are now addressed in the Discussion. We are only reporting a correlate, and we have not determined whether our observed neurons contribute to fear output. The question concerning the amygdala-vlPAG relationship is a valid one. We favor an interpretation in which central amygdala inputs train up the vlPAG, and with sufficient training vlPAG activity becomes independent. In support, Ozawa and colleagues have shown that inhibiting CeA inputs to the vlPAG has no impact on activity at cue onset and only partially reducing vlPAG ramping activity (Ozawa et al., 2017). Further, Lee and colleagues have shown that the CeA is required for the acquisition of conditioned suppression with minimal training trials, but that with more extensive fear conditioning, the CeA is no longer required (Lee et al., 2005). I observed a similar pattern for the CeA in one of my dissertation experiments (McDannald, 2010). This discussion is now included in the Discussion.

2) Further, it is worth noting that uncertainty in and of itself is a stressor, which could explain the fact that the fear output towards the uncertainty cue is higher than might be expected. It could be that the amygdala is encoding the threat probability and that information is being conveyed to the vlPAG, but that the additional stress from uncertainty is being encoded elsewhere in the brain and added to the fear output response at some unknown mediating output structure.

We agree and data from our laboratory may have identified such a signal that reflects threat probability plus additional stress from uncertainty: the retrorubral field. We discuss this possibility but keep it brief, as it is highly speculative (Discussion).

3) In the Results it would be really helpful to see how many rats/neurons etc. were included in the study as I had to search through to find this information.

We have gone through the manuscript to include more instances of specifically stating the number of rats recorded from and the numbers included in each population. We have also explicitly stated that the number of Onset and Ramping neurons from each subject is shown in full in the supplementary figures that include all waveforms.

4) In Figure 1—figure supplement 3, the authors suggest (at least as I understand it) that there is no relation between normalised firing rate and suppression ratio in the overall population of neurons, but from looking at the graph it appears that there might be at least a weak negative correlation? That is, when normalised firing rate was lowest, in the danger condition, suppression ratio appears to be at its highest (indicating high fear), and vice versa for the safety conditions. Is that correct or am I missing something?

We included Figure 1—figure supplement 3 so that neural activity, and two indicators of fear output (nose poke rate and suppression ratio), could be visualized in once place. From our view, the main takeaways are (1) the two measures of fear output strongly resemble one another. This means that by using a ratio, we weren’t fundamentally changing the data. (2) Behavior was very consistent across all recording sessions. That is, the behavior in sessions from which we recorded Onset neurons (neurons 1-29 on y axis) was virtually identical to the other recording sessions. This means that the correlate we observed was not due to sampling odd or outlier behavioral sessions. (3) It is also apparent from this figure that almost no neurons are uniquely safety-responsive.

5) In the last paragraph of the subsection “Ramping neurons signal threat probability through delay until shock receipt”, Figures 1E and 1F are referred to, I believe this should say Figures 6E and 6F.

The error has been corrected.